# Photochemical degradation of iron(III)-citrate/citric acid aerosol quantified with the combination of three complementary experimental techniques and a kinetic process model

Jing Dou[1], Peter A. Alpert[2], Pablo Corral Arroyo[2*], Beiping Luo[1], Frederic Schneider[2], Jacinta Xto[3], Thomas Huthwelker[3], Camelia N. Borca[3], Katja D. Henzler[3], Jörg Raabe[4], Benjamin Watts[4], Hartmut Herrmann[5], Thomas Peter[1], Markus Ammann[2], and Ulrich K. Krieger[1]

[1]Institute for Atmospheric and Climate Science, ETH Zürich, 8092 Zürich, Switzerland
[2]Laboratory of Environmental Chemistry, Paul Scherrer Institute, 5232 Villigen, Switzerland
[3]Laboratory for Synchrotron Radiation and Femtochemistry, Paul Scherrer Institute, 5232 Villigen, Switzerland
[4]Laboratory for Synchrotron Radiation-Condensed Matter, Paul Scherrer Institute, 5232 Villigen, Switzerland
[5]Atmospheric Chemistry Department (ACD), Leibniz-Institute for Tropospheric Research (TROPOS), 04318 Leipzig, Germany
[*]now at: Department of Chemistry and Applied Biosciences, ETH Zürich, 8093 Zürich, Switzerland

**Correspondence:** Jing Dou (jing.dou@env.ethz.ch) and Ulrich K.Krieger (ulrich.krieger@env.ethz.ch)

**Abstract.**

Iron(III) carboxylate photochemistry plays an important role in aerosol aging, especially in the lower troposphere. These complexes can absorb light over a broad wavelength range, inducing the reduction of iron(III) and the oxidation of carboxylate ligands. In the presence of $O_2$, ensuing radical chemistry leads to further decarboxylation, and the production of $\cdot OH$, $HO_2$, peroxides, and oxygenated volatile organic compounds, contributing to particle mass loss. The $\cdot OH$, $HO_2$, and peroxides in turn re-oxidize iron(II) back to iron(III), closing a photocatalytic cycle. This cycle is repeated resulting in continual mass loss due to the release of $CO_2$ and other volatile compounds. In a cold and/or dry atmosphere, organic aerosol particles tend to attain highly viscous states. While the impact of reduced mobility of aerosol constituents on dark chemical reactions has received substantial attention, studies on the effect of high viscosity on photochemical processes are scarce. Here, we choose iron(III)-citrate ($Fe^{III}(Cit)$) as a model light absorbing iron carboxylate complex that induces citric acid (CA) degradation to investigate how transport limitations influence photochemical processes. Three complementary experimental approaches were used to investigate kinetic transport limitations. The mass loss of single, levitated particles was measured with an electrodynamic balance, the oxidation state of deposited particles was measured with X-ray spectromicroscopy, and $HO_2$ radical production and release into the gas phase was observed in coated wall flow tube experiments. We observed significant photochemical degradation, with up to 80 % mass loss within 24 hours of light exposure. Interestingly, we also observed that mass loss always accelerated during irradiation, resulting in an increase of the mass loss rate by about a factor of 10. When we increased relative humidity, the observed particle mass loss rate also increased. This is consistent with strong kinetic transport limitations for highly viscous particles. To quantitatively compare these experiments and determine important physical and chemical parameters, a numerical multi-layered photochemical reaction and diffusion (PRAD) model that treats chemical reactions and transport of various species was developed. The PRAD model was tuned to simultaneously reproduce all experimental results

as closely as possible and captured the essential chemistry and transport during irradiation. In particular, the photolysis rate of $Fe^{III}$, the re-oxidation rate of $Fe^{II}$, $HO_2$ production, and the diffusivity of $O_2$ in aqueous $Fe^{III}(Cit)$/CA system as function of relative humidity and $Fe^{III}(Cit)$/CA molar ratio could be constrained. This led to satisfactory agreement within model uncertainty for most, but not all experiments performed. Photochemical degradation under atmospheric conditions predicted by the PRAD model shows that release of $CO_2$ and re-partitioning of organic compounds to the gas phase may be very significant to accurately predict organic aerosol aging processes.

## 1 Introduction

Photochemistry in the atmosphere (either in the gas phase or in the particle phase) plays an important role in aerosol aging processes. Photochemically produced free radicals in the gas phase (mainly $\cdot$OH) can be taken up by aerosol particles, inducing multi-phase chemistry. However, uptake is limited by the collision rate and condensed phase molecular transport when diffusion coefficients are sufficiently low, which restricts chemical reactions to the near-surface region of the particle. In contrast, photochemically generated radicals in aerosol particles would be present throughout their bulk due to light penetrating their whole volume. Direct photochemical reaction induced radical production occurs when the energy of light quanta is high enough (mostly the UV part of the solar spectrum) to cause bond cleavage or rearrangement in a molecule. In the lower troposphere where UV light intensity is low, indirect photochemistry initiated by visible radiation may become significant. Important indirect photochemical processes are transition metal complex photochemistry and photosensitized processes (Corral Arroyo et al., 2018; George et al., 2015). This work focuses on iron carboxylate catalyzed photochemistry due to its abundance and reactivity in the atmosphere.

Iron is the most abundant transition metal in the earth's crust. Wind erosion is the main source of iron in the atmosphere, but anthropogenic activities such as industrial processes, traffic and combustion processes can also potentially release iron in particulate form (Deguillaume et al., 2005). Depending on parameters such as temperature, pH value, ionic strength and concentration of involved substances, iron can combine with inorganic or organic ligands to form complexes (Deguillaume et al., 2005; Faust and Hoigné, 1990; Kieber et al., 2005). Iron can be found complexed with low molecular weight inorganic species such as the hydroxide anion ($OH^-$), sulfate ($SO_4^{2-}$) and sulfite ($SO_3^{2-}$) (Brandt and van Eldik, 1995; Hofmann et al., 1991; Weschler et al., 1986). Quantifying iron atmospheric processing and solubility is of global importance, especially for nutrient input into the Worlds oceans (Hamilton et al., 2019; Kanakidou et al., 2018). Heterogeneous chemistry involving particulate iron and $SO_2$ can result in sulfate formation and increase aerosol loading (Grgić et al., 1998, 1999; Grgiè, 2009). Additionally, iron photochemical processing in aerosol particles, fog droplets and cloud water is an important radical source (Bianco et al., 2020; Abida et al., 2012) and sink for organic compounds (Weller et al., 2014, 2013; Herrmann et al., 2015). Organic compounds are a major component in atmospheric aerosol particles and have received more and more attention as potential ligands for iron(III) complexation. For instance, humic-like substances derived from water soluble organic compounds, have been reported to be strong chelating ligands with iron(III) (Dou et al., 2015; Kieber et al., 2003; Okochi and Brimblecombe, 2002; Willey et al., 2000). Oxalate and other carboxylates have been identified to be important ligands for iron(III) because they

are available in sufficient amounts (Chebbi and Carlier, 1996; Kahnt et al., 2014; Kawamura et al., 1985), and the carboxylate groups are acidic enough to dissociate and chelate with iron(III) at atmospheric pH values (Okochi and Brimblecombe, 2002). Field studies have confirmed that soluble iron is mostly in complexes with carboxylate functions (Tapparo et al., 2020; Tao and Murphy, 2019).

In atmospheric aqueous phases, iron normally exists in oxidation states (II) and (III), and they can convert into each other via redox cycling. The ratio between iron(II) and iron(III) in aerosol particles is quite variable, which depends on several factors such as the presence of light, oxidizing compounds, and ligands. For example, Grgić et al. (1999) reported that the concentration ratio of Fe(II) to Fe(III) varied between 0.9 and 3.1 in urban aerosol particles with size range $0.4 - 1.6\ \mu m$. Iron(III) carboxylate complexes $[Fe^{III}(OOC-R)]^{2+}$ are well-known photoactive compounds (Wang et al., 2012; Weller et al., 2013, 2014). They can easily get excited by light in the UV-VIS range, inducing ligand-to-metal charge transfer (LMCT) (Cieśla et al., 2004), which is an inner sphere electron transfer (i.e., the electron transfer occurs via a covalently bound bridging ligand) from the carboxylate group to the iron. Investigations using time-resolved transient spectroscopy reported the formation of long lived radical complexes, $[Fe^{II}(\cdot OOC-R)]^{2+}$, with lifetimes of the order of a millisecond, followed by the dissociation to the organic radical $R-COO\cdot$ and an $Fe^{II}$ aquacomplex (Feng et al., 2007; Glebov et al., 2011; Pozdnyakov et al., 2009; Zhang et al., 2009):

$$[Fe^{III}(OOC-R)]^{2+} + h\nu \rightarrow [Fe^{III}(OOC-R)]^{2+*} \tag{R1}$$

$$[Fe^{III}(OOC-R)]^{2+*} \rightarrow [Fe^{II}(\cdot OOC-R)]^{2+} \tag{R2}$$

$$[Fe^{II}(\cdot OOC-R)]^{2+} \rightarrow Fe^{2+} + R-COO\cdot \tag{R3}$$

$R-COO\cdot$ will decarboxylate almost instantaneously ($k_{R3} \approx 10^9 - 10^{12}\ s^{-1}$) (Abel et al., 2003; Bockman et al., 1997; Hilborn and Pincock, 1991):

$$R-COO\cdot \rightarrow R\cdot + CO_2 \tag{R4}$$

The alkyl radical $R\cdot$ will react rapidly with dissolved $O_2$, producing a peroxy radical with $k_{R4} \approx 2 \times 10^9\ M^{-1}\ s^{-1}$ (von Sonntag and Schuchmann, 1991):

$$R\cdot + O_2 \rightarrow RO_2^\cdot \tag{R5}$$

Subsequent reactions of $R\cdot$ and $RO_2^\cdot$ are specific depending on the type of ligand and its substitution.

In this work we investigated iron(III)-citrate ($[Fe^{III}(OOCCH_2)_2C(OH)(COO)]$, in short $Fe^{III}(Cit)$), as a model species to better understand iron carboxylate photochemistry in atmospheric aerosol particles. Our $Fe^{III}(Cit)$ system undergoes LMCT

reaction in the same way as countless other iron(III)-carboxylate compounds (Cieśla et al., 2004; Weller et al., 2013, 2014). Its photochemical reaction scheme is well established in both solution (Abida et al., 2012; Faust and Zepp, 1993; Pozdnyakov

et al., 2012) and solid states (Abrahamson et al., 1994). Citric acid (CA) is an established proxy for oxygenated atmospheric organic matter, with its thermodynamic properties, water diffusivity and viscosity being well studied (Lienhard et al., 2012, 2014; Song et al., 2016). For these reasons, it is a valid and reliable proxy for atmospheric iron-carboxylate photochemical processes.

As schematically described in Fig. 1, $Fe^{III}$(Cit) absorbs light up to 500 nm, inducing LMCT, followed by immediate de-

carboxylation of the central carboxyl group, since the hydroxyl group adjacent to a carboxyl group facilitates decarboxylation (Weller et al., 2013). In the presence of $O_2$, oxidants such as $HO_2$ and $H_2O_2$ will be produced, which can oxidize $Fe^{II}$ back to $Fe^{III}$ via Fenton reactions (Fenton, 1894), with additional production of oxidants. $Fe^{III}$ then combines with another citric acid in this aqueous system, closing the photocatalytic cycle, in which iron acts as a catalyst for CA degradation. In addition, the generation of reactive oxygen species (ROS) and peroxy radicals leads to further decarboxylation and more production of

oxygenated volatile organic compounds (OVOCs) (e.g., acetone) (Pozdnyakov et al., 2008; Wang et al., 2012). Therefore, this photodegradation process is potentially an important sink of carboxylate groups in the troposphere.

We expect that continuing chemistry subsequent to initial photochemical reaction steps in the aerosol phase will be significantly altered by diffusion limitations when $Fe^{III}$(Cit) particles mixed with CA attain a high viscosity. As viscosity increases, molecular diffusion coefficients tend to decrease (Koop et al., 2011) and therefore, photochemical cycling will also be slow.

Increasing water content is expected when RH increases and will effectively plasticize particles (Koop et al., 2011) leading to a more well mixed conditions and faster photochemical cycling when compared with lower RH. However, these effects have been investigated in dark systems but not in photochemical systems (Berkemeier et al., 2016; Shiraiwa et al., 2011; Shiraiwa and Seinfeld, 2012; Steimer et al., 2015a). In order to better understand this system and how it reacts to RH, we used a triad of photochemical experiments including electrodynamic balance (EDB), scanning transmission X-ray microscopy coupled with

near edge absorption fine structure (STXM/NEXAFS) spectroscopy, and a coated wall flow tube (CWFT) to investigate how particle size, mass, and indicators of chemical composition change during photochemical processes. In this work we mostly focus on the humidity dependence of this photochemical degradation, while Alpert et al. (2020, under review) focus on the impacts on ROS species and the fate of free radicals during this photochemical degradation.

To perform quantitative comparison of these experiments and determination of relevant properties, a numerical multi-layered

photochemical reaction and diffusion (PRAD) model that treats chemical reactions and transport of various species was developed. In addition, we will use the PRAD model to simulate photochemical aging processes under atmospheric conditions. In the following we briefly discuss our experimental approaches in Sect. 2, and include a detailed explanation of the PRAD model in section 2.5. A comparison between experimental results and the PRAD model is presented in Sect. 3. Finally, we discuss the impact and atmospheric importance of kinetic limitations to photochemical degradation in Sect. 4.

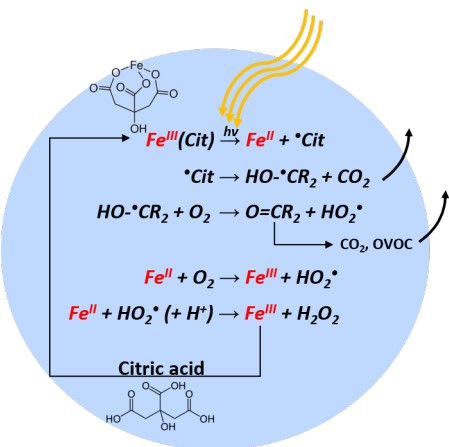

**Figure 1.** Photocatalytic cycle of $Fe^{III}(Cit)$ complex in an aqueous particle containing citric acid, with explicit charge balance given in Table 2. $R$ corresponds to the carboxylate side chain $-CH_2COO^-$.

## 2 Methods

### 2.1 Solution preparation

Citric acid ($\geq 99.5$ %) and Iron(III) citrate tribasic monohydrate ($18 - 20$ % Fe basis) were purchased from Sigma-Aldrich. Iron(II) citrate ($Fe^{II}(HCit)$) was purchased from Dr. Paul Lohmann GmbH KG. Dilute aqueous solutions of $Fe^{III}(Cit)$/citric acid and $Fe^{II}(HCit)$/citric acid were made in ultrapure water ($18\,\mathrm{M\,\Omega\,cm^{-1}}$, MilliQ). Since $Fe^{III}(Cit)$ only dissolves slowly in water, citric acid solutions with suspended $Fe^{III}(Cit)$ crystals inside were sonicated for at least 24 hours and the same dissolving procedure was also applied to the $Fe^{II}(HCit)$ powders. Note, that all the procedures were done under red light illumination because $Fe^{III}(Cit)$ is light sensitive. The molar ratio between $Fe^{III}(Cit)$ and CA was different for each experimental method used in this study. For EDB, STXM/NEXAFS and CWFT experiments, stock solutions were prepared with molar ratios of 0.05, 1.0 and 0.07, respectively.

### 2.2 Bulk property measurements by EDB

We used an electrodynamic balance (EDB) to measure the mass loss in single, levitated particles under irradiation. The experimental setup has been described previously (Steimer et al., 2015b). In short, an electrically charged aqueous particle (radius $\sim 10\,\mathrm{\mu m}$) is injected into an EDB. The balance is of the double ring design (Davis et al., 1990) with a high AC voltage applied to the two-parallel electrode rings and a DC voltage across hyperbolic endcaps. The DC field compensates the gravitational force of the particle and is used as a measure for the mass of the particle. The EDB is placed in a three wall glass chamber, with a cooling liquid (ethanol) pumped through the two inner walls and an insulation vacuum between two outer walls, to control the temperature (T) at the location where the particle levitates. The relative humidity (RH) within the chamber is regulated by

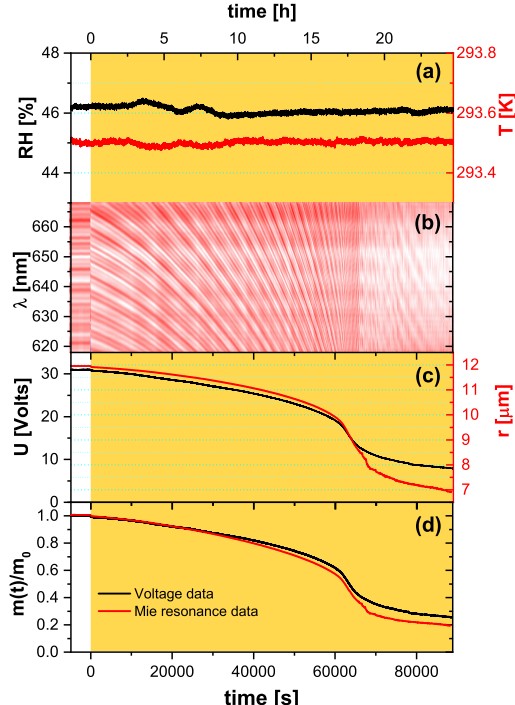

**Figure 2.** Raw EDB experimental data for a $Fe^{III}(Cit)$/citric acid (molar ratio of 0.05) particle. (a) Relative humidity (left axis, black) and temperature (right axis, red) of the droplet. (b) Intensity map of low resolution Mie resonance spectra. (c) Black line: DC voltage that compensates gravitational force; Red line: radius of the droplet, with the initial value of the particle radius determined using high resolution Mie resonance spectroscopy (not shown), and the change in radius determined from low resolution Mie resonance data given in (b). (d) Normalized mass remaining ratio deduced from DC voltage (black), and from Mie resonance shift (red) - assuming constant density. Yellow shaded region in panels (a), (c), and (d) indicate the time of laser irradiation with $0.25 \ W \ cm^{-2}$ at 375 nm.

adjusting the ratio of a dry and humidified gas flow through the chamber. In the experiments described in this work, we used a typical total flow of 40 sccm and set the total pressure inside the cell at $8 \times 10^4 \ Pa$.

The spherical particles were characterized by two Mie resonance spectroscopy based methods: (i) A narrow bandwidth tunable diode laser (TDL, tuning range $765-781$ nm) was used to determine the refractive index and radius simultaneously with high precision (Steimer et al., 2015b). (ii) Simultaneously, a broad-band LED centered around 640 nm was used to illuminate the particle. And the backscatter signal from the LED is recorded using a spectrograph with a slow scan back-illuminated CCD (charge-coupled device) array detector, to follow the resonance wavelength shift of the particle (Zardini et al., 2006). If

we assume refractive index stays constant during the experiment, the radius change of a particle is easily calculated from this resonance wavelength shift:

$$\frac{r(t)}{r_0} = \frac{\lambda_0 + \Delta\lambda(t)}{\lambda_0} = 1 + \frac{\Delta\lambda(t)}{\lambda_0}. \tag{1}$$

We illuminated particles to induce photochemical reaction with either a cw diode laser emitting at 375 nm (LuxX 375-20, Omicron Laserage) or a frequency doubled diode laser emitting at 473 nm (gem 473, Laser Quantum). At the wavelength of 375 and 473 nm, $Fe^{III}(Cit)$ is reported to have a molar absorptivity of 796 and 60.7 $M^{-1}cm^{-1}$, respectively (Pozdnyakov et al., 2008).

In a typical EDB experiment, we let the particle equilibrate to $RH$ and $T$ in a pure $O_2$ gas phase for up to 10 hours in the dark before irradiation. Exemplary raw data of an experiment at 46 % RH and 293.5 K is shown in Fig. 2. The measured DC voltage compensating the gravitational force, as well as the radius of the particle deduced from Mie-resonance spectroscopy decreased dramatically during illumination in the first 18 hours, with more than half of the initial mass lost to the gas phase. Note, that the radius and mass loss rates increased as seen in Fig. 2(c). We assumed refractive index and density of the particle did not change upon photochemistry and therefore, the mass loss calculated from the DC voltage could be directly compared with size change by calculating the particle mass remaining ratio,

$$\frac{m(t)}{m_0} = \left\{ \frac{r(t)}{r_0} \right\}^3, \tag{2}$$

where $m_0$ is the particle mass prior to irradiation. Mass loss derived from both ways independently is shown in Fig. 2(d) and reveal that there is a little difference between the mass loss up to $t \approx 65000$ s, corresponding to when $\frac{m(t)}{m_0} < 0.4$. Therefore, the refractive index and density are mostly governed by those of aqueous citric acid up until the half the particle mass is lost. The total mass loss over 24 hours irradiation is more significant and drops by 80 % for the particular experiment shown in Fig. 2. In addition, we observed the mass loss rate initially was $\sim 1.3$ % $h^{-1}$ and increased to $\sim 14$ % $h^{-1}$ when 40 % to 60 % of the initial mass was lost. This mass loss acceleration is discussed further in detail with the help of the PRAD model simulations in the section 3.1. At $t \approx 65000$ s, the mass loss slowed down considerably when we observed a distortion in the Mie-resonance pattern (Fig. 2(b) and video in the Supplement). The distortion may be attributable to partial crystallization of iron citrate in the particle, which would explain the slowing photochemical degradation.

## 2.3 Chemical characterization by STXM/NEXAFS

STXM/NEXAFS measurements were performed at the PolLux endstation located at the Swiss Light Source (SLS) to obtain the Fe oxidation state of particles between 0.2-2 µm in diameter (Flechsig et al., 2007; Frommherz et al., 2010; Raabe et al., 2008). Particles containing $Fe^{III}(Cit)$/CA were nebulized from aqueous solution with a mole ratio between $Fe^{III}(Cit)$:CA of 1:1. They were dried in air at $RH < 30$ % and impacted onto silicon nitride membranes mounted in portable sample holders. The sample holders were transported to the endstation in an evacuated container and shielded against ambient light. Once there, they were mounted in the PolLux environmental microreactor (Huthwelker et al., 2010), and kept under a total pressure of $1.5 \times 10^4$ Pa, $T = 293.5K$, $RH = 40, 50$ or $60$ %, with a controlled gas flow. Further details of sample preparation are provided in previous literature (Alpert et al., 2019; Huthwelker et al., 2010; Steimer et al., 2014). The microreactor was mounted into a vacuum chamber for in situ STXM/NEXAFS analysis. When desired, the microreactor could operate in vacuum conditions without a gas flow. The transmission of X-ray photons through the particles were measured and converted to optical density, $OD = -\ln(I/I_0)$, where $I$ and $I_0$ are the transmitted and incident photon flux as a function of X-ray energy. The Fe L-edge

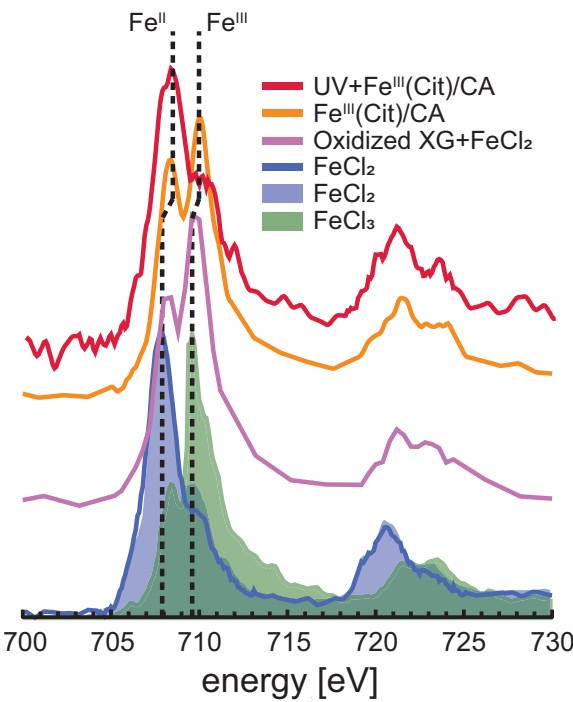

**Figure 3.** Iron L-edge NEXAFS spectra of $Fe^{III}(Cit)$/CA particles before and after irradiation with UV light shown as orange and red, respectively. The previously recorded spectrum from mixed xanthan gum (XG) and $FeCl_2$ particles exposed to ozone is shown as the purple line, and a spectrum from $FeCl_2$ particles is shown as the blue line (Alpert et al., 2019). $FeCl_2$ and $FeCl_3$ spectra from Moffet et al. (2012) are shown as the blue and green shading, respectively. The vertical dashed lines indicate peak X-ray absorption at 707.9 shifted to 708.3 eV for $Fe^{II}$ and 709.6 eV shifted to 710.0 eV for $Fe^{III}$.

absorption was probed over the X-ray energy range of $700 - 735$ eV. Figure 3 shows an example of NEXAFS spectra of $Fe^{III}(Cit)$/CA particles before (orange) and after (red) irradiation with UV light. X-ray energy calibration was consistently performed using $FeCl_2$ and compared with previous literature for $FeCl_2$ and $FeCl_3$ salts (Moffet et al., 2012) and a mixture of xanthan gum and $FeCl_2$ oxidized by $O_3$ (Alpert et al., 2019). The peak absorption for iron(II) and iron(III) are at X-ray

energies of 708.3 and 710.0 eV. We were capable of resolving peaks separated by 0.4 eV at the Fe L-edge. Ferrous and ferric iron peaks are separated by 1.7 eV and thus, clearly distinguishable. Following a previous procedure (Alpert et al., 2019), we imaged particles at these two energies to determine the OD ratio between them. Then the fraction of $Fe^{III}$ out of total Fe, $\beta$, was determined using the parameterization from Moffet et al. (2012). It is important to note that the X-ray energies absorption peaks observed for $FeCl_2$ and $FeCl_3$ were identical for $FeCl_2$ mixed with xanthan gum either unexposed or exposed to $O_3$ (Alpert

et al., 2019). However, we have found that these peaks shifted by about $+0.4$ eV, possibly due to the strong complexation with CA. Small shifts in energy can occur depending on the chemical environment surrounding Fe atoms (Garvie et al., 1994; Moffet et al., 2012). In agreement with Alpert et al. (2019), the peak absorption energies for our particles were independent of $RH$ from 0 % to 60 % within $\pm 0.2$ eV. When calculating $\beta$, we always imaged particles at at 708.3 and 710.0 eV.

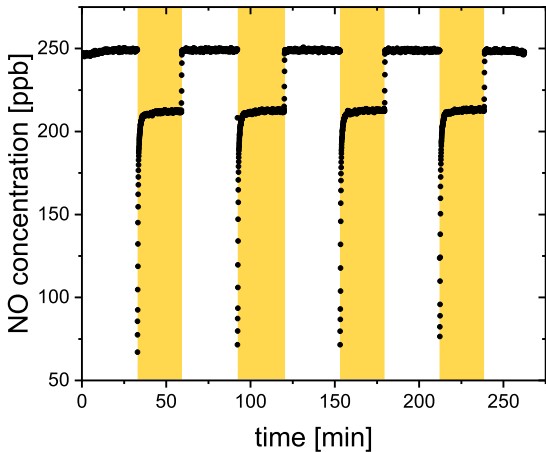

**Figure 4.** NO concentration raw data from CWFT film containing $Fe^{III}(Cit)/CA$ (molar ratio of 0.07) with lamps on (yellow shaded region) and off at 29.3 % RH and 298.15 K.

## 2.4 HO$_2$ production determined by CWFT

The HO$_2$ release upon irradiation of a $Fe^{III}(Cit)/CA$ thin film was measured by scavenging HO$_2$ with an excess of NO in a coated wall flow tube (CWFT) reactor (Duran glass, inside diameter 1.2 cm, long 50 cm). The film was composed of $Fe^{III}(Cit)/CA$, and deposited inside the tubular glass flow tube with a thickness between $0.15 - 0.2$ µm and an error of about 20 %. Details of the film preparation are described previously (Corral Arroyo et al., 2018; González Palacios et al., 2016). Seven UV lamps (UV-A range, Philips Cleo Effect) were mounted surrounding the glass reactor held at 298.15 K.

The total light output between $300 - 590$ nm was 210 W m$^{-2}$. The flows of N$_2$, O$_2$, and NO passing through the reactor were controlled. The NO concentration during CWFT photochemical experiments was in excess ($> 10^{13}$ molecules cm$^{-3}$) to efficiently scavenge 99 % of HO$_2$ produced by the film. The concentration of NO was tracked by a chemiluminescence detector (Ecophysics CLD 77 AM). In an example CWFT experiment at $RH = 29.3$ %, a clear NO loss was observed when UV lights were switched on as shown in Fig. 4, which was due to the release of HO$_2$ radicals into the gas phase and reaction

of NO with HO$_2$ forming NO$_2$ and OH$\cdot$. OH$\cdot$ is then scavenged by NO producing HONO. The production of HONO was routinely checked as described in González Palacios et al. (2016). Therefore, the production rate of HO$_2$, $P_{HO_2}$, was calculated from the loss rate of NO assuming a 2:1 ratio to HO$_2$ conversion:

$$P_{HO_2} = \frac{[NO] \times \text{flow}}{2 S_{film}}, \tag{3}$$

where [NO] is the loss of gas-phase concentration of NO in molecules cm$^{-3}$, flow is the volumetric gas flow in the CWFT in

205    cm$^3$ s$^{-1}$, and S$_{film}$ is the surface area of the film in cm$^2$.

## 2.5 Development of the PRAD model

We developed a photochemical reaction and diffusion (PRAD) model to interpret our experiments and to understand any feedbacks between transport limitations and photochemistry, especially under low $RH$ conditions, corresponding to high viscosity of the particle phase. The PRAD model consists of two modules: a detailed chemical process module, treating equilibria and chemical reactions, and a transport module handling the physical transport of all species (including diffusion in the aqueous phase as well as gas-particle phase partitioning). Conceptually, the PRAD model relies on the kinetic model framework for aerosol surface chemistry and gas-particle interactions (Pöschl et al., 2007), similar as for example the KM-GAP model (Shiraiwa et al., 2012). Numerically, the PRAD model uses a Euler forward step method as explained in detail below, while KM-Gap solves coupled differential equations. In passing, there are alternative approaches, for example Kinetiscope (Houle et al., 2015) does not integrate sets of coupled differential equations to predict the time history of a chemical system. Instead, it uses a general stochastic algorithm to propagate a reaction.

As illustrated in Fig. 5, the PRAD model divides a spherical droplet into a number of shells, $n$, which exchange molecules after each chemical time step. Shell thickness and numbers of shells were adjusted to enable the resolution of steep concentration gradients within a reasonable computation time. The volume of each shell was constant instead of the thickness and the shells become thinner and thinner from the center to the surface of the particle. For each shell and at each time step, we first calculated the composition using the thermodynamic equilibria of the $Fe^{III}(Cit)$/CA system, as listed in Table 2. Then the Newton-Raphson method (Burden and Faires, 2011) was used to calculate the turnover and the concentration of products and reactants over time for the chemical reactions listed also in Table 2 with a fixed time step of 0.2 s. After each time step diffusion of all species between the shells and the evaporation of products (or condensation of the reactant $O_2$) were computed. The time step, $\Delta t$, for physical transport processes was determined dynamically to ensure both numerical stability and computational efficiency.

For each species, the molar flux from shell $i$ to the next shell $i+1$ was calculated as

$$f_i = -4\pi r_i^2 D_l \frac{dc}{dr}\bigg|_{r=r_i} = -4\pi r_i^2 D_l \frac{c_{i+1} - c_i}{0.5(r_{i+1} - r_{i-1})}; \qquad \forall i \in \{1, 2, ..., n-1\}, \tag{4}$$

where $D_l$ is the liquid phase diffusion coefficient of the corresponding species. Shell $i$ extends from $r_{i-1}$ to $r_i$, while shell $i+1$ extends from $r_i$ to $r_{i+1}$, with $r$ being the distance from the particle center. In Eq. (4), $c$ is the molar concentration in each shell of the aqueous particle, defined as

$$c_i = \frac{N_i}{V_i}; \qquad \forall i \in \{1, 2, ..., n\}, \tag{5}$$

where $N_i$ is moles of a particular species in shell $i$, and $V_i$ is the total volume of shell i.

At the outermost shell $n$, the gas-particle phase partitioning of each species was determined by the modified Raoult's law. The flux from shell $n$ into the gas phase, $f_n$, was calculated to be

$$f_i = -4\pi r_n^2 D_g \frac{dc}{dr}\bigg|_{r=r_n} = -4\pi r_n^2 D_l \frac{c_g - c_g^*}{r_n} = -4\pi r_n D_g \frac{p_{partial} - p_{vapor}}{RT}; \tag{6}$$

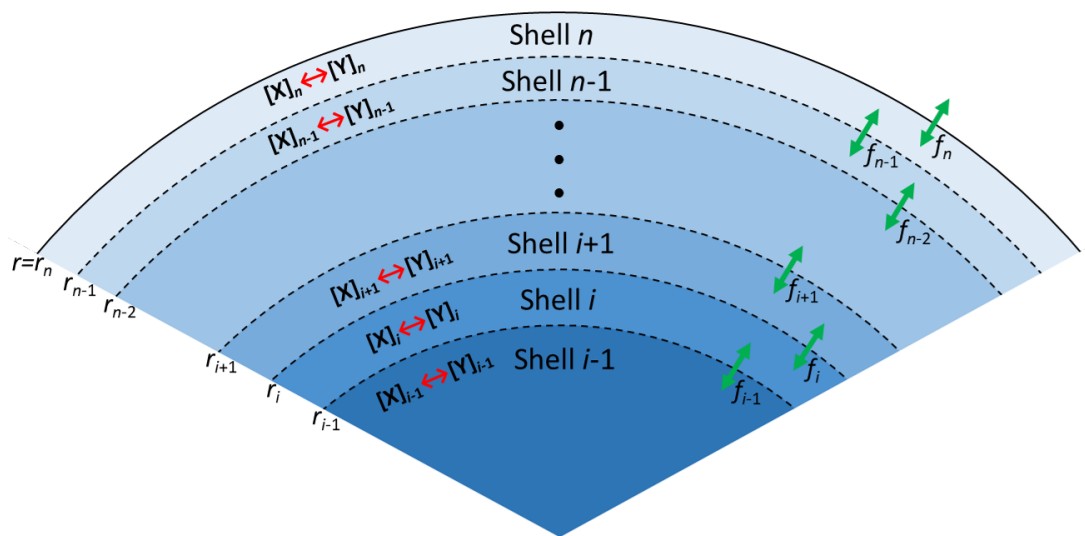

**Figure 5.** Schematic illustration of the PRAD model. Shells, transport fluxes (green arrows), and chemical processes (red arrows) of each species. The particle is radially symmetric with the surface of the particle marked as $r_n$.

where $D_g$ is the gas phase diffusion coefficient of the corresponding species, and $R$ is the gas constant. $p_{\mathrm{partial}}$ is the partial pressure of the species in the gas phase at infinite distance from the particle. Based on Henry's law, vapor pressure of the species $p_{\mathrm{vapor}}$ is defined as

$$p_{\mathrm{vapor}} = \frac{c_n}{H}, \tag{7}$$

where $c_n$ is the molar concentration in the outermost shell $n$, and $H$ is the Henry's law coefficient. Substituting Eq. (7) into Eq. (6) leads to

$$f_n = -4\pi r_n D_g \left( \frac{p_{\mathrm{partial}}}{RT} - \frac{c_n}{HRT} \right). \tag{8}$$

We calculated the partial pressure of $H_2O$ from $RH$, and took the partial pressure of $O_2$ based on the total pressure and the fraction of $O_2$ gas flow used in experiments. For other species, $p_{\mathrm{partial}}$ is negligible and assumed to be zero. So for all species other than $H_2O$ and $O_2$, $f_n$ simplifies to

$$f_n = -4\pi r_n D_g \left( 0 - \frac{c_n}{HRT} \right) = 4\pi r_n D_g \frac{c_n}{HRT}. \tag{9}$$

Based on the calculated fluxes, the change in moles, $\Delta N_i$, of each species in $\Delta t$ was given by

$$\Delta N_i = (f_{i-1} - f_i)\Delta t; \qquad \forall i \in \{1, 2, ..., n\}, \tag{10}$$

from which the concentration and corresponding shell thickness using the molar volume of each species were recalculated for the next time step.

The chemical loss rate of $O_2$ was treated in the liquid phase diffusion module instead of in the chemical module, since the loss can be very fast and its life time can be smaller than $10^{-6}$ s. If $O_2$ loss due to reaction was determined in the chemical module, a chemical time step of $10^{-6}$ s would result in extensive computational time beyond what is reasonable for this study. Therefore, within each liquid phase diffusion time step, in addition to the physical transport, the chemical loss of $O_2$ molecules was calculated in each shell

$$\frac{dN_i}{dt} = -k_{O_2} N_i; \qquad \forall i \in \{1, 2, ..., n\}. \tag{11}$$

Integration yields

$$N_i = N_i e^{-k_{O_2} \Delta t}; \qquad \forall i \in \{1, 2, ..., n-1\}, \tag{12}$$

where $k_{O_2}$ is total chemical reaction rate of $O_2$. In the outermost shell $n$, $O_2$ molar concentration is assumed to be at steady state all times, meaning that the chemical loss of $O_2$ is always compensated by the condensation of $O_2$ from the gas phase:

$$-k_{O_2} N_{n,O_2} = -4\pi r_n D_g \left( \frac{p_{\text{partial},O_2}}{RT} - \frac{c_n}{H_{O_2} RT} \right). \tag{13}$$

Substituting Eq. (5) in Eq. (13), we calculate the moles of $O_2$ in this shell, $N_{n,O_2}$, as

$$N_{n,O_2} = \frac{p_{\text{partial},O_2}}{k_{O_2} RT / (4\pi r_n D_g) + 1/(H_{O_2} V_n)}. \tag{14}$$

where $H_{O_2}$ is Henry's law coefficient of $O_2$.

All iron containing compounds and all free ions are considered not to evaporate from the particle phase. All other species have Henry's law coefficients given in Table 2. If the evaporation rate of some species is fast enough, then their concentration in the outermost shell $n$ can always be at steady state, which means

$$f_{n-1} = f_n, \tag{15}$$

that is,

$$-4\pi r_{n-1}^2 D_l \frac{c_n - c_{n-1}}{0.5(r_n - r_{n-2})} = 4\pi r_n D_g \frac{c_n}{HRT}. \tag{16}$$

From Eq. (16), $c_n$ can be deduced to be

$$c_n = c_{n-1} \frac{1}{1 + 0.5(r_n - r_{n-2}) r_n D_g / (HRT r_{n-1}^2 D_l)}. \tag{17}$$

And the number of moles of the volatile species in the outermost shell $N_n$ as

$$N_n = c_n V_n. \tag{18}$$

We have parameterized aqueous and gas phase diffusion coefficients, $D_l^j$ and $D_g^j$, respectively, for all species $j$ given in Appendix A1 and A2. There are several other assumptions and approximations made to adapt this model to the aqueous $Fe^{III}(Cit)/CA$ system:

1. We set water activity in the particle phase always in equilibrium with the gas phase, since the RH of the gas phase did not change during each experiment.

2. The bulk accommodation coefficients of all species were assumed to be 1.

3. Neglecting the influence of $Fe^{III}(Cit)$, the water activity was taken from that of CA ($a_w^{CA}$), which has been determined from the mass fraction of CA, $W_{CA}$, by Lienhard et al. (2012)

$$a_w^{CA} = \frac{1 - W_{CA}}{1 + q \cdot W_{CA} + r \cdot W_{CA}^2}, \tag{19}$$

with

$$q = -3.16761 + 0.01939T - 4.02725 \times 10^{-5}T^2, \tag{20}$$

and

$$r = 6.59108 - 0.05294T + 1.06028 \times 10^{-4}T^2, \tag{21}$$

The water activity of citrate ($a_w^{Cit}$) was calculated using the same equation

$$a_w^{Cit} = \frac{1 - W_{Cit}}{1 + q \cdot W_{Cit} + r \cdot W_{Cit}^2}, \tag{22}$$

where the mass fraction of citrate $W_{Cit}$ was treated in two fractions: citrate without Fe and citrate containing Fe. The water activity of an aqueous 1 M $Fe^{III}(Cit)$ solution was determined using a water activity meter (AquaLab water, Model 3B, Decadon Device, USA) at room temperature. We found this water activity corresponds to that of a 0.81 M aqueous CA solution. Therefore, for calculating water activity, all iron containing citrate complexes (no matter $Fe^{II}$ or $Fe^{III}$) were treated with a reduced concentration: namely with a factor of 0.81 of the corresponding citrate molarity. Hence, the overall amount of citrate was calculated as

$$N_{Cit}^* = N_{Cit} + 0.81 N_{FeCit}, \tag{23}$$

accordingly and $W_{Cit}$ was determined following

$$W_{Cit} = \frac{M_{Cit} \cdot N_{Cit}^*}{M_{Cit} \cdot N_{Cit}^* + M_{H_2O} \cdot N_{H_2O}}, \tag{24}$$

where $M_{Cit}$ and $M_{H_2O}$ are the molar mass of CA and water, respectively. For other species $j$, the contribution to the water activity is proportional to their molar volume ($MV_j$), so that in total,

$$a_w = a_w^{CA} \times a_w^{Cit} \times \frac{MV_{H_2O}}{MV_{H_2O} + \sum_j MV_j}. \tag{25}$$

4. $Fe^{III}(Cit)$ photolysis, decarboxylation and oxidation of the alcohol group in presence of $O_2$ yields the compounds $O=C(CH_2COO)_2^{2-}$ or $O=C(CH_2COOH)_2$, which are $C_5$ species. We assumed that half of $C_5$ species undergoes photochemical reactions to produce $CO_2$ and compounds with $2 - 4$ carbon atoms, $C_4$, $C_3$ and $C_2$ (see reactions R10−R14 shown in Table 2), all of which are capable of being released to the gas phased depending on their solubility.

5. We estimated the quantum yield in reactions R1 and R2 in Table 2, as $\Phi = 1.0$ at $\lambda = 375$ nm and $\Phi = 0.002$ at $\lambda = 473$ nm (Dou et al., 2019), and we parameterized $\Phi$ as a function of wavelength, $\lambda$:

$$\Phi = \frac{e^{-0.145(\lambda-430)}}{1+e^{-0.145(\lambda-430)}}. \tag{26}$$

In total, the PRAD model includes 13 equilibria and 17 chemical reactions among 32 species, as well as their condensed phase diffusivities and Henry's law coefficients. Some of these parameters are known from previous studies (see Tables 1 and 2 for references), while others are not known and difficult to estimate. For instance, even though absorption spectra of $Fe^{III}(Cit)$ has been measured in aqueous solution (Pozdnyakov et al., 2012), the corresponding quantum yield has not, which leaves the photolysis rate of $Fe^{III}(Cit)$, $j$, unknown. Also, there are no data reported of the diffusivity of $O_2$ in aqueous citric acid solutions, and the chemical reaction rate of the oxidation of the $Fe^{II}$-citrate complex by $O_2$ is quite uncertain (Gonzalez et al., 2017). In order to find the optimal parameter set, we compared experimental data of the three setups taken under well-controlled conditions with model predictions and tuned the unknown parameters manually.

We restricted our tuning of the parameters to reach satisfactory agreement with all experimental data simultaneously. The equilibrium constants and rate coefficients that were tuned are indicated in Table 2 (the sensitivity of the PRAD model results to a few of its parameters is shown in appendix A5). The parameters were adjusted in a wide and acceptable range until a good representation of our data could be obtained. For example, the fraction of iron(III) in a photoactive complex (equilibirum E5 in Table 2) must have been high enough to reproduce STXM/NEXAFS observations that iron could be reduced to low levels as seen in Fig. 7 described later. In comparison, E7 must have been much lower than E5 so that the amount of iron(III) in a non-photoactive complex was small compared to being in complex with citrate. As another example, oxidation of $Fe^{2+}$ (R5-R8 in Table 2) is fairly well-referenced, and therefore, we adjusted the rate of reaction R9 until the model reoxidation rates matched those observed. Tuning of individual bulk diffusion coefficients for all species was not attempted. Instead, we simplified the representation of diffusion coefficients using a parameterization as function of molar mass described in appendix A1. The 2 constants in Eqn (A8) and 2 constants in Eqn (A3) were tuned resulting in the absolute diffusion coefficients shown in Fig. A1. Henry's law coefficients for gasses were tuned, however purposefully set at values higher than expected for pure water or highly dilute aqueous solution. This was inspired by previous studies regularly reporting solubility of, e.g. $O_2$ and $CO_2$ higher in a variety organic liquids than water (Fogg, 1992; Battino et al., 1983). It is important to note that the result of this tuning does not mean that we found the global minimum in the parameter space, see e.g. (Berkemeier et al., 2017). A thorough search for a global minimum for our model with 16 tuning parameters for chemistry, 4 tuning parameters (and our parameterization) for diffusion and 9 tuning parameters for solubility is computationally very expensive and beyond the scope of this paper. However, for our purpose here, namely modeling typical timescales of photochemical degradation of organic aerosol under atmospheric conditions (see Sec. 3.5), the PRAD model framework should allow sufficiently accurate predictions. In other words, we expect similar mass degredation in atmospheric particles due to the fact that many other relevant iron-carboxylate compounds undergo LMCT similarly as to our model system (Weller et al., 2013, 2014). Additionally, if a particular system requires parameter values that significantly differ than ours, the PRAD model framework itself should still be valid. Note, that

careful evaluation is needed when picking a single parameter of the PRAD model for use in another context. Comparison of the refined model with our experimental data are shown in the next section.

**Table 1.** Liquid phase diffusivity factors (normalized to water) and Henry's law coefficients (Sander, 2015) of major species in $Fe^{III}(Cit)$ photochemistry system.

| number | name | formula | $l_f^j$ † | $H_0$ ($M\,atm^{-1}$) ‡ | $Q$ ‡ |
|---|---|---|---|---|---|
| 1 | water | $H_2O$ | 1 | 1 | 1 |
| 2 | cit total | - | - | - | - |
| 3 | ferric ($Fe^{III}$) total | - | - | - | - |
| 4 | ferrous ($Fe^{II}$) total | - | - | - | - |
| 5 | citric acid (CA) | $(CH_2COOH)_2C(OH)(COOH)$ /$H_3Cit$ | $1.20 \times 10^{-6}$ | infinite | 10000 |
| 6 | dihydrogen citrate | $(CH_2COOH)_2C(OH)(COO)^-$ /$H_2Cit^-$ | $1.20 \times 10^{-6}$ | infinite | 10000 |
| 7 | hydrogen citrate | $(CH_2COOH)C(OH)(CH_2COO)(COO)^{2-}$ /$HCit^{2-}$ | $1.20 \times 10^{-6}$ | infinite | 10000 |
| 8 | citrate | $C(OH)(CH_2COO)_2(COO)^{3-}$ /$Cit^{3-}$ | $1.20 \times 10^{-6}$ | infinite | 10000 |
| 9 | | $Fe^{III}(Cit)(OH)^-$ | $3.92 \times 10^{-7}$ | infinite | 10000 |
| 10 | | $Fe^{III}(HCit)^+$ | $5.04 \times 10^{-7}$ | infinite | 10000 |
| 11 | ferrous citrate | $Fe^{II}(HCit)$ | $5.04 \times 10^{-7}$ | infinite | 10000 |
| 12 | ferric citrate | $Fe^{III}(Cit)$ | $5.04 \times 10^{-7}$ | infinite | 10000 |
| 13 | ferric ion | $Fe^{3+}$ | $3.78 \times 10^{-5}$ | infinite | 10000 |
| 14 | | $Fe^{III}(OH)^{2+}$ | $1.18 \times 10^{-5}$ | infinite | 10000 |
| 15 | ferrous ion | $Fe^{2+}$ | $3.78 \times 10^{-5}$ | infinite | 10000 |
| 16 | hydrogen ion | $H^+$ | - | - | - |
| 17 | hydroxide ion | $OH^-$ | - | - | - |
| 18 | hydroperoxyl radical | $HO_2^{\cdot}$ | $1.13 \times 10^{-4}$ | $4 \times 10^4$ | 5900 |
| 19 | superoxide radical | $O_2^{\cdot-}$ | - | - | - |
| 20 | radicals | $OH - \cdot C(CH_2COO)_2^{2-} + OH - \cdot C(CH_2COOH)_2$ | $2.69 \times 10^{-6}$ | infinite | 10000 |
| 21 | | | | | |
| 22 | hydroxyl radical | $\cdot OH$ | - | - | - |
| 23 | hydrogen peroxide | $H_2O_2$ | $1.07 \times 10^{-4}$ | $8.3 \times 10^5$ | 7400 |
| 24 | oxygen | $O_2$ | $1.20 \times 10^{-4}$ | $3.5 \times 10^{-2}$ | 1500 |
| 25 | carbon dioxide | $CO_2$ | depends on $a_w$ and $T$ | $3.4 \times 10^{-1}$ | 2400 |
| 26 | acetone | $CH_3COCH_3$ | $1.18 \times 10^{-5}$ | 30 | 4600 |
| 27 | unk prod $C_4$ | $C_4$ | $5.30 \times 10^{-6}$ | $1 \times 10^5$ | 6000 |
| 28 | | $O=C(CH_2COO)_2^{2-} + O=C(CH_2COOH)_2$ | $1.95 \times 10^{-6}$ | infinite | 8000 |
| 29 | acetic acid | $CH_3COOH$ | $3.23 \times 10^{-5}$ | $4.1 \times 10^3$ | 6300 |
| 30 | | $Fe^{II}[O=C(CH_2COO)_2]$ | $7.22 \times 10^{-7}$ | infinite | 10000 |
| 31 | unk prod $C_5$ | $C_5$ | $2.69 \times 10^{-6}$ | $1 \times 10^7$ | 8000 |
| 32 | unk prod $C_5$ | $C_5\_stable$ | $2.69 \times 10^{-6}$ | $5 \times 10^8$ | 8000 |

† $l_f^j$ is a factor of the diffusion coefficient of each species $j$ normalized to that of water.

‡ Henry's law is described as a function of temperature T: $H = H_0 e^{\frac{Q}{T} - \frac{Q}{T_0}}$.

**Table 2.** Compilation of equilibria, chemical reactions, and corresponding rate constants in $Fe^{III}(Cit)$ photochemistry system.

| number | reactions | $K_{eq}/k_r/\sigma$ | sources |
|---|---|---|---|
| E1 | $H_2O \rightleftharpoons OH^- + H^+$ | $1 \times 10^{-14}$ M | |
| E2 | $H_3Cit \rightleftharpoons H_2Cit^- + H^+$ | $7.5 \times 10^{-4}$ M | Martell and Smith (1982) |
| E3 | $H_2Cit^- \rightleftharpoons HCit^{2-} + H^+$ | $1.7 \times 10^{-5}$ M | Martell and Smith (1982) |
| E4 | $HCit^{2-} \rightleftharpoons Cit^{3-} + H^+$ | $4.0 \times 10^{-7}$ M | Martell and Smith (1982) |
| E5 | $Fe^{3+} + Cit^{3-} \rightleftharpoons Fe^{III}(Cit)$ | $1.58 \times 10^{13}$ M$^{-1}$ | tuning parameter |
| E6 | $Fe^{3+} + Cit^{3-} + H_2O \rightleftharpoons Fe^{III}(Cit)(OH)- + H^+$ | $8.35 \times 10^7$ M$^{-1}$ | tuning parameter |
| E7 | $Fe^{3+} + HCit^{2-} \rightleftharpoons Fe^{III}(HCit)^+$ | $2.51 \times 10^7$ M$^{-1}$ | tuning parameter |
| E8 | $Fe^{2+} + HCit^{2-} \rightleftharpoons Fe^{II}(HCit)$ | $1.935 \times 10^{10}$ M$^{-1}$ | tuning parameter |
| E9 | $Fe^{3+} + H_2O \rightleftharpoons Fe^{III}(OH)^{2+} + H^+$ | $4.57 \times 10^{-3}$ M | Smith and Martell (1976) |
| E10 | $O_2^{\cdot -} + H^+ \rightleftharpoons HO_2^{\cdot}$ | $6.3 \times 10^4$ M$^{-1}$ | Bielski et al. (1985) |
| E11 | $Fe^{2+} + O{=}C(CH_2COO)_2^{2-} \rightleftharpoons Fe^{II}[O{=}C(CH_2COO)_2]$ | $2 \times 10^3$ M$^{-1}$ | tuning parameter |
| E12 | $2H^+ + OH-\cdot C(CH_2COO)_2^{2-} \rightleftharpoons OH-\cdot C(CH_2COOH)_2$ | $1.5 \times 10^6$ M$^{-2}$ | tuning parameter |
| E13 | $2H^+ + O{=}C(CH_2COO)_2^{2-} \rightleftharpoons O{=}C(CH_2COOH)_2$ | $1.5 \times 10^6$ M$^{-2}$ | tuning parameter |
| R1 | $Fe^{III}(Cit) + h\nu \rightarrow Fe^{2+} + OH-\cdot C(CH_2COO)_2^{2-} + CO_2$ | $3.0 \times 10^{-18}$ (at 375 nm) or $2.3 \times 10^{-19}$ (at 473 nm) cm$^2$ | Pozdnyakov et al. (2012) |
| R2 | $Fe^{III}(Cit)(OH)-\cdot + h\nu \rightarrow Fe^{2+} + OH-\cdot C(CH_2COO)_2^{2-} + OH^- + CO_2$ | | |
| R3 | $OH-\cdot C(CH_2COO)_2^{2-} + O_2 \rightarrow O{=}C(CH_2COO)_2^{2-} + O_2^{\cdot -} + H^+$ | $1 \times 10^6$ M$^{-1}$ s$^{-1}$ | Hug et al. (2001) |
| R4 | $HO_2 + HO_2 \rightarrow H_2O_2 + O_2$ | depends on $a_w$ | tuning parameter |
| R5 | $Fe^{2+} + O_2^{\cdot -} (+2H^+) \rightarrow Fe^{3+} + H_2O_2$ | $1 \times 10^7$ M$^{-1}$ s$^{-1}$ | Rush and Bielski (1985) |
| R6 | $Fe^{2+} + HO_2^{\cdot}(+H^+) \rightarrow Fe^{3+} + H_2O_2$ | $1.2 \times 10^6$ M$^{-1}$ s$^{-1}$ | Rush and Bielski (1985) |
| R7 | $Fe^{2+} + H_2O_2 \rightarrow Fe^{3+} + \cdot OH + OH^-$ | $76$ M$^{-1}$ s$^{-1}$ | Walling (1975) |
| R8 | $Fe^{2+} + \cdot OH \rightarrow Fe^{III}(OH)^{2+}$ | $4.3 \times 10^8$ M$^{-1}$ s$^{-1}$ | Christensen and Sehested (1981) |
| R9 | $Fe^{II}(HCit) + O_2 \rightarrow Fe^{III}(Cit) + HO_2^{\cdot}$ | $0.05$ M$^{-1}$ s$^{-1}$ | tuning parameter |
| R10 | $C_5 + h\nu \rightarrow C_3 + 2CO_2$ | $1 \times 10^{-21}$ cm$^2$ | tuning parameter |
| R11 | $C_4 + h\nu \rightarrow C_3 + CO_2$ | $1 \times 10^{-20}$ cm$^2$ | tuning parameter |
| R12 | $C_5 + h\nu \rightarrow C_2 + C_3$ | $1 \times 10^{-22}$ cm$^2$ | tuning parameter |
| R13 | $C_4 + h\nu \rightarrow C_2 + C_2$ | $1 \times 10^{-21}$ cm$^2$ | tuning parameter |
| R14 | $C_5 + h\nu \rightarrow C_4 + CO_2$ | $1 \times 10^{-20}$ cm$^2$ | tuning parameter |
| R15 | $C_5$_stable fraction | $0.5$ | tuning parameter |
| R16 | radical self reaction | $0$ | tuning parameter |

## 3 Comparisons between experimental measurements and model simulations

### 3.1 The effect of $RH$ on photocatalytic degradation efficiency

We performed experiments with single, levitated particles under continuous UV irradiation (375 nm) in pure $O_2$ at different $RH$ to access the effects of $RH$ on the photocatalytic cycle shown in Fig. 1, and tested the model performance under these conditions. Qualitatively, a continuous decrease of particle mass and size is expected to occur due to evaporation of volatile products, as shown in Fig. 2. Figure 6 shows the fraction of particle mass remaining with the irradiation time at three different $RH$, calculated from resonance wavelength shifts (Eqs. (1) and (2)). Clearly, particle mass was lost to the gas phase with time due to the evaporation of photochemical products and similar to Fig. 2, all data show a very significant acceleration of mass loss with time. After tuning some of the parameters of the model as discussed further below, the PRAD model simulations reproduces our data with a very similar trend and magnitude over all, which gives us confidence that the PRAD model captures the essential chemistry and transport during irradiation. (How particle mass evolves subsequently until 80 % mass loss in both experiments and models is shown in Fig. A4.) However, the model is not able to capture the full degree of acceleration of the degradation rate, as it does not attempt to include the complete multi-generational oxidation chemistry at the level of individual components after initial radical production. The degradation processes were faster at higher RH. At lower RH, the particle was expected to be more viscous, diffusion coefficients were expected to be lower, products were generated at a lower rate, and volatile products moved more slowly to the surface to evaporate. More importantly, $O_2$ taken up by the particle from the gas phase diffused more slowly into the bulk of the particle at lower RH, thus less $HO_2$ and $H_2O_2$ formed and less $Fe^{II}$ could be re-oxidized from the surface to the center of the particle. The observed gradient in $Fe^{III}$ fraction, $\beta$, and the modelled gradients in $O_2$ and ROS in the particle have been shown with radial profiles in Alpert et al. (2020, under review). This resulted in fewer photochemically active $Fe^{III}$ complexes available for photocatalytic degradation. The characteristic degradation time shortened by a factor of 5.5 when $RH$ increased from 46 % and 61 %, which demonstrates that photochemical cycling is highly sensitive to the microphysical conditions. The diffusivity of $O_2$ must have significantly impacted re-oxidation reaction rates. In addition, the diffusion coefficients of both $Fe^{II}$ and $Fe^{III}$ species increases with RH. Therefore, the molecular transport between both iron and oxygen reactants increases causing a highly non-linear trend in increasing mass loss with increasing $RH$.

### 3.2 Determination of iron(III) reduction rate and iron(II) re-oxidation rate by STXM/NEXAFS

In STXM/NEXAFS experiments, the freshly prepared $Fe^{III}(Cit)$ mixed with CA at $x = 1.0$ particles were irradiated to determine the $Fe^{III}(Cit)$ photolysis rate, as shown in Fig. 7. Each experimental data point is the average $Fe^{III}$ fraction from $16 - 36$ individual particles. Fitting an exponential function, $\beta = \beta_0 e^{j_{obs}t}$, yields $\beta_0 = 0.93 \pm 0.09$ and a first order decay rate of $j_{obs} = 0.08 \pm 0.01 \text{ s}^{-1}$. The LED power at the sample was measured to be $5.9 \pm 0.6 \text{ mW}$ in total and had a Gaussian spectral profile between $361 - 374 \text{ nm}$ at full width-half max. When mounting the UV fiber optics and collimator lens for multiple samples, the illuminated area had a circle equivalent diameter of $5 \pm 1.5 \text{ mm}$. Using the absorption cross section calculated from the molar attenuation coefficient (Pozdnyakov et al., 2008), $\Phi = 1.0$ and propogating all uncertainties yields a photochemical reaction rate of $j_{calc} = 0.20 \pm 0.12 \text{ s}^{-1}$, which is in agreement with $j_{obs}$. This implies that assuming a quantum yield of 1 at

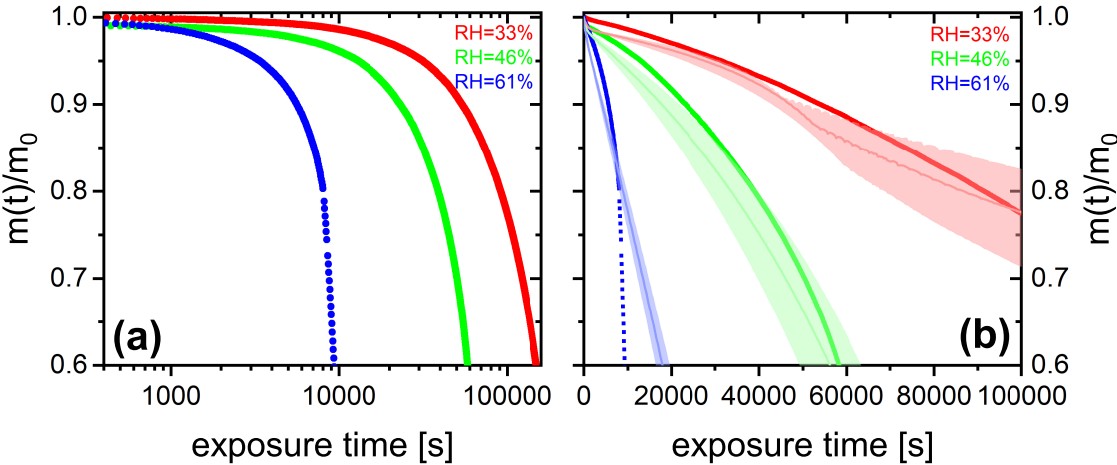

**Figure 6.** (a) $Fe^{III}(Cit)$/CA (molar ratio 0.05) particle mass change with irradiation time (log scale) at different RH: 33 % (red), 46 % (green), and 61 % (blue). The irradiation wavelength was 375 nm, its intensity was 0.25 W cm$^{-2}$, and the experimental temperature was 293.5 K. (b) EDB experimental data in (a) with PRAD outputs at corresponding RH (with $\pm 2$ % RH uncertainty shown as shaded area) as a function of irradiation time (linear scale).

these UV wavelengths is reasonable. This calculated value for the photochemical reaction rate was used in the PRAD model for analyzing the STXM/NEXAFS experiments shown in Fig. 7. We note that our estimate of the photochemical reduction rate and agreement with data is based on the reproducibility of setting up our optical system in the X-ray vacuum chamber. In Fig. 8 (discussed below), this setup procedure was performed for RH=40% and RH=50-60% independently, and still, iron reduction was in agreement with model predictions in the first minutes after UV light was switched off. The width of the red shading in Fig. 7 is large, and likely the UV-fiber setup was the largest source of error for $j$. A better estimate would require repeat measurements as a function of RH e.g. to elucidate any systematic uncertainty on iron reduction reactions due to viscosity changes. However, this was not possible as usage of the X-ray beam for STXM/NEXAFS experiments was limited to a few days to complete all experiments.

In a different set of experiments, we irradiated particles using much lower power setting having $j = 2.2 \times 10^{-3}$ s$^{-1}$ in a mixed He and $O_2$ atmosphere and at a fixed RH for 15 min to reduce $Fe^{III}$ to $Fe^{II}$. The UV light was then switched off to allow re-oxidation in the dark while measuring $\beta$ over time. Figure 8 shows $\beta$ as a function of time at $RH = 40, 50,$ and 60 %. Clearly, the $Fe^{III}$ fraction increased significantly slower with time at drier conditions. While particles were observed to re-oxidize to 70 % within 2 hours at 60 % RH and expected to be completely re-oxidized within about 6 hours according to PRAD model simulations, no significant re-oxidation occurred on this timescale for the particles exposed to only 40 % RH. Modelling the re-oxidation with the PRAD model yields very satisfactory agreement, indicating that the diffusivity parameterizations of the model are capturing the $RH$ dependence of the molecular transport in the viscous matrix.

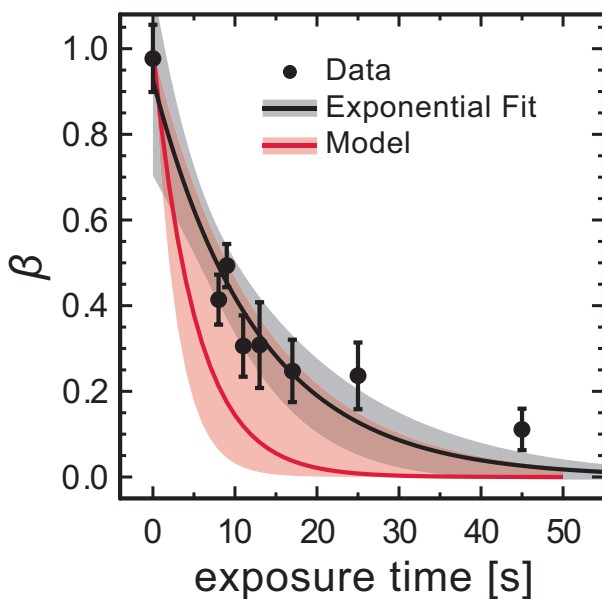

**Figure 7.** Loss of $Fe^{III}$ in $Fe^{III}(Cit)$/CA (molar ratio of 1) particles as a function of light exposure time determined using STXM/NEXAFS at 293.5 K. Each data point is the average $Fe^{III}$ fraction over about $16 - 36$ individual particles. An exponential fit, $\beta = \beta_0 e^{j_{obs}t}$, yields for the initial condition is $\beta_0 = 0.93 \pm 0.09$ and a first order decay rate of $j_{obs} = 0.08 \pm 0.01 \text{ s}^{-1}$. The black shading indicates 95 % confidence on the exponential fit. The PRAD model prediction and uncertainty are given as red solid line and shading, respectively, and uses a calculated decay rate of $j_{calc} = 0.20 \pm 0.12 \text{ s}^{-1}$.

### 3.3 Determination of iron(II) re-oxidation rate with single, levitated particle using EDB

As the experiments with single, levitated particles yield only bulk properties and not the oxidation state of iron-citrate directly, we designed a dedicated experimental procedure to indirectly determine the re-oxidation rate of $Fe^{II}$. We used multiple irra-
395 diation and re-oxidation repetitions as shown schematically in Fig. 9. Initially, we exposed a newly injected $Fe^{III}(Cit)$/CA aqueous particle to blue laser irradiation (473 nm, 4 W cm$^{-2}$) in pure $N_2$ for 500 s to ensure all $Fe^{III}$ was reduced through photolysis reactions R1 and R2 (listed in Table 2). As previously described, these reactions led to $CO_2$ production with subsequently loss of $CO_2$ to the gas phase, which was observed as a shift in the Mie-resonance wavelength. This shift is shown in Fig. 2(b) and on an enlarged scale in Fig. 10. After irradiation, we switched the gas flow from $N_2$ to $O_2$ in the dark, and
400 $Fe^{II}$ was oxidized back to $Fe^{III}$ over time in this period, either by ROS (R5$-$R8) or directly by $O_2$ (R9). After a desired time spent in $O_2$, the gas flow was switched back to $N_2$ followed by irradiation to repeat the photolysis step done initially. The ratio of the Mie-resonance wavelength shift of the two photolysis steps was set proportional to the ratio of re-oxidized $Fe^{III}/Fe_{tot}$. These two steps (i.e., photolysis in $N_2$ and re-oxidation in $O_2$) were repeated several times, but between each irradiation the particle was exposed to $O_2$ for different time periods. Following this procedure we intended to map out the characteristic time
for re-oxidation at various $RH$.

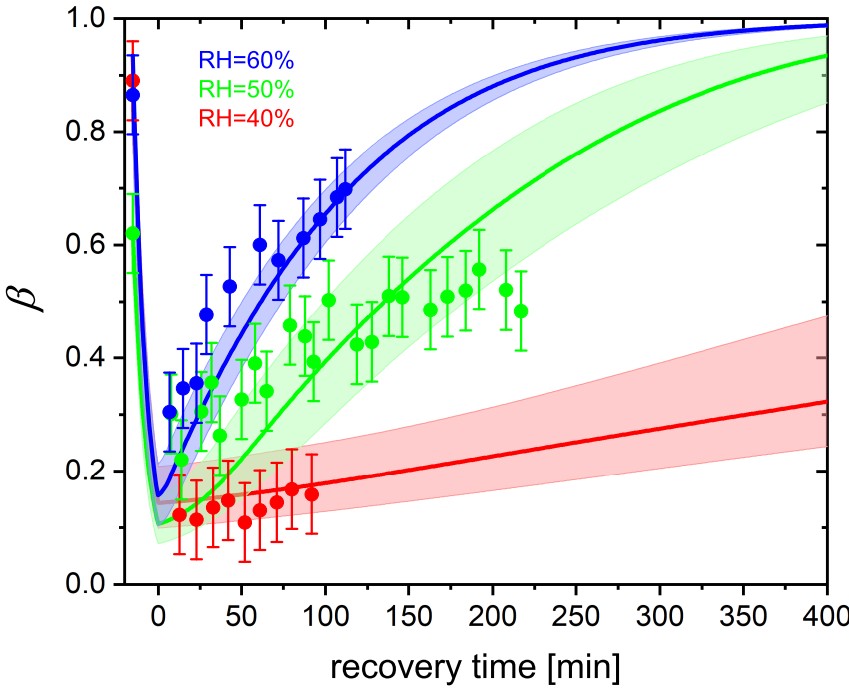

**Figure 8.** Re-oxidation of $Fe^{III}$ as a function of exposure time in $O_2$ from STXM observations. Time before 0 represents initial 15 min irradiation procedure under He. $Fe^{III}(Cit)/CA$ (molar ratio of 1) particles at 293.5 K with 40 % RH (red dots), 50 % RH (green dots), and 60 % RH (blue dots). Lines: red (40 % RH), green (48 % RH), and blue (65 % RH) are the $Fe^{III}$ fractions predicted using the PRAD model, the shaded areas indicate model output assuming $\pm 2$ % RH, $\pm 0.07$ initial $Fe^{III}$ fraction, and $\pm 1.8$ % light intensity uncertainty in the STXM experimental conditions.

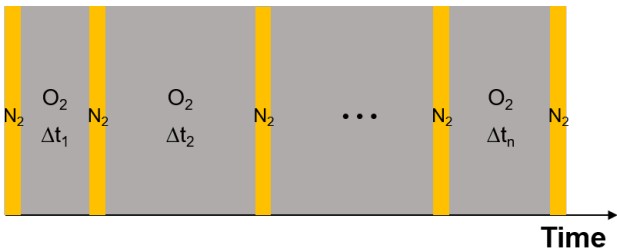

**Figure 9.** Schematic of the experimental procedure to investigate the re-oxidation rate of $Fe^{II}$. Orange columns represent laser irradiation (473 nm, 4 W cm$^{-2}$), each irradiation takes place in pure $N_2$ for a period of 500 s; grey columns mark the recovery process in pure $O_2$ in the dark, here the time interval is varied. For details see text.

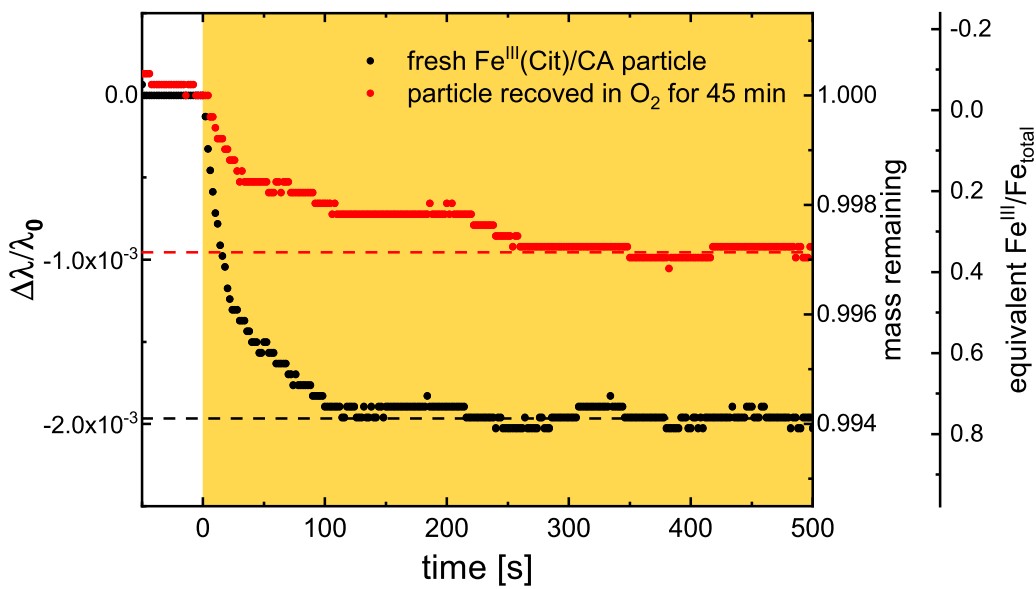

**Figure 10.** Temporal evolution of two Mie resonance wavelengths shifts and particle mass changes normalized to the wavelength and mass prior irradiation; the experiment was done at 48 % RH and 293.5 K. The orange area marks the time with irradiation (473 nm, 4 W cm$^{-2}$). Black: first irradiation with a fresh Fe$^{III}$(Cit)/CA particle in N$_2$, red: irradiation in N$_2$ right after the particle recovered in O$_2$ for 45 min in the dark. Two horizontal dashed lines are indicating the final mass remaining after CO$_2$ loss to the gas phase. For explanation of scaling of the third axis, indicating the equivalent Fe$^{III}$/Fe$_{tot}$ ratio, see text.

An example of the corresponding raw data (293.5 K and 48 % RH) and retrieved oxidation state is shown Fig. 10. Black circles indicate the first irradiation step with a fresh Fe$^{III}$(Cit)/CA particle, and red circles indicate the irradiation step that followed the particle after exposure to O$_2$ for 45 min in the dark. Clearly, the resonance wavelength decreased more during the first irradiation than the second. Therefore, we can conclude unambiguously that the Fe$^{III}$(Cit)/CA particle initially had

more Fe$^{III}$ than what could be re-oxidized in O$_2$ for 45 min. Quantitative scaling however, requires knowledge of the initial Fe$^{III}$ fraction. Our experiments showed that long exposure (tens of hours) to O$_2$ yielded larger Mie-resonance shifts than those of the initial photolysis of the freshly prepared particle. This indicated that the initial Fe$^{III}$ fraction was less than 1.0. Hence, we normalized the Fe$^{III}$ fraction accordingly to the data at long ($>$ 15 hours) exposure times. For the experiment included in Fig. 10 for example, the initial Fe$^{III}$ fraction of the particle was 0.76, indicating that the particle has been partially reduced

during sample preparation. After the particle has been totally photoreduced, exposure to O$_2$ for 45 min did not re-oxidize all reduced Fe$^{II}$ to Fe$^{III}$, but only 0.36 Fe$^{III}$ has been recovered (as shown with two horizontal dashed lines in Fig. 10).

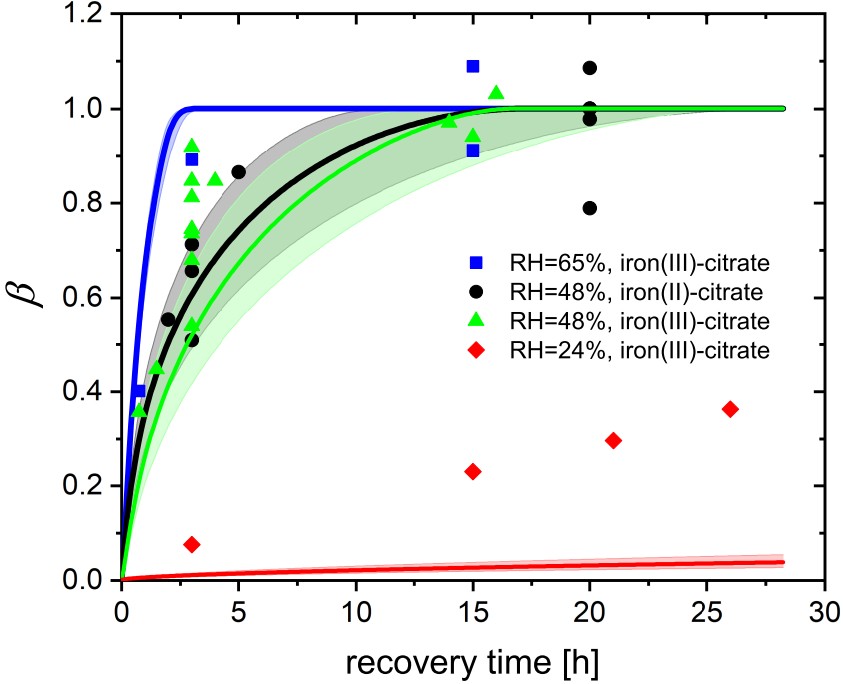

**Figure 11.** $Fe^{III}$ fraction after different time intervals of recovery in $O_2$ from EDB observations. $Fe^{III}(Cit)/CA$ (molar ratio of 0.05) particles at 293.5 K with 24 % RH (red diamonds), 48 % RH (green triangles), and 65 % RH (blue squares). Black dots: experiments with $Fe^{II}(HCit)/CA$ particles at 48 % RH. Lines: red (24 % RH), black (48 % RH), and blue (65 % RH) are the $Fe^{III}$ fractions predicted using the PRAD model, the shaded areas indicate model output assuming $\pm 2$ % RH uncertainty in the EDB experimental conditions.

Another set of experiments was done by starting with freshly injected $Fe^{II}(HCit)/CA$ particles instead of $Fe^{III}(Cit)/CA$. The only difference in experimental procedure, compared to what is described above, is that there is no first irradiation step. Instead, the particle is initially exposed to $O_2$ in the dark for a certain time interval, during which $Fe^{II}$ is oxidized only directly 420 by $O_2$ (reaction R9). Afterwards, the same irradiation and recovery procedures as reported above were taken. The equivalent $Fe^{III}$ fractions of all experiments are shown in Fig. 11. Any uncertainty in the normalization of an individual experiment will cause a corresponding uncertainty in the normalized mass loss, which made estimating the uncertainty for individual data points impossible. Nevertheless, from the complete data set, it was evident that with longer time intervals in $O_2$, more $Fe^{III}$ was recovered. At 48 % and 65 % RH, about 10 hours and 3 hours exposure to $O_2$, respectively, was sufficient for all $Fe^{II}$ 425 to be re-oxidized, while at 24 % RH, the recovery even after 25 h was not yet complete. The general trend is consistent with our observation in the STXM/NEXAFS experiments (section. 3.2) and can be attributed to molecular diffusion limitations at lower RH: it takes more time for $O_2$ to diffuse into the particle, and for $Fe^{II}$ to diffuse out to the surface of the particle to react

with $O_2$. It was also evident from these data that $Fe^{II}$ re-oxidized by $O_2$ is as important as $Fe^{II}$ re-oxidized by radicals and peroxides as there is no significant difference between the experiments starting from $Fe^{II}(HCit)$/CA compared to those with $Fe^{III}(Cit)$/CA. In addition, it indicates that both Fe(II) and Fe(III) can act as a photocatalyst as long as Fe(II) can be oxidized to Fe(III), which was also confirmed by Grgić et al. (1999).

By tuning the direct oxidation rate of $Fe^{II}(HCit)$ by $O_2$ (R9) and the diffusivity of $O_2$, we are able to model the recovery rate at different $RH$ using the PRAD model as shown in Fig. 11. There is satisfactory agreement for the larger $RH$, but significant underestimation of re-oxidation for the experiments at 24 % RH. Our model requires the reaction rate coefficient of R9 to be $0.05$ $M^{-1} s^{-1}$, which is a factor of 60 smaller than the value that Gonzalez et al. (2017) estimated from their model ($3 \pm 0.7$ $M^{-1} s^{-1}$). The liquid phase diffusivity of $O_2$ in our model is $7.1 \times 10^{-19}$ $m^2 s^{-1}$ at 24 % RH, $1.9 \times 10^{-15}$ $m^2 s^{-1}$ at 48 %, and $2.1 \times 10^{-14}$ $m^2 s^{-1}$ at 65 % RH. These diffusivities of $O_2$ are $2-4$ orders of magnitude smaller than those of $CO_2$ determined in Dou et al. (2019). However, we need to stress that in the model, some of the iron related complex equilibrium constants and their diffusion coefficients, and the Henry's law coefficient of $O_2$ at different water activities are highly uncertain as well, yielding to a significant uncertainty in the determination of $O_2$ diffusivity. For example, if the solubility of $O_2$ would be less than what our parameters predict now, a larger $O_2$ diffusivity would be consistent with our data. In addition, with the total gas flow used in our experiments, it takes about 8 min for replacing the entire EDB gas volume from $N_2$ to $O_2$, and from the response of the particle to flow condition change, we estimate an interval of about 30 min to reach full equilibration to the new gas phase conditions. Therefore, the life time of organic radicals needs to be reconsidered. In the PRAD model, we do not take the radical-radical self reactions (R16) into account, which may turn out to be a significant sink for the radicals. However, it should be pointed out that the parameter set we have now is a good compromise with additional constraints from STXM/NEXAFS and CWFT experiments.

Another approximate approach to analyse the data of Fig. 11 is to use the analytical solutions for a reacto-diffusive kinetic regime. Here, $O_2$ taken up from the gas phase by a particle remains confined to a very thin layer below its surface compared to its size provided it reacts reasonably fast with the organic components. Under these conditions there are always pairs of reaction rate and diffusion constants representing the experiments equally well (Alpert et al., 2019; Steimer et al., 2014). In the reacto-diffusive framework, with constrained reaction rate and Henry's law coefficient of $O_2$, the diffusion coefficient of $O_2$ can be estimated to be $3.6 \times 10^{-16}$ and $4.4 \times 10^{-15}$ $m^2 s^{-1}$ at 48 % and 65 % RH, respectively (details are given in Appendix A4). These are both one order of magnitude less than the values from PRAD model prediction, but still consistent with each other when considering all uncertainties.

### 3.4   $HO_2$ production measured by CWFT experiments

The CWFT experiment allows us to investigate another aspect of the photochemistry of the $Fe^{III}(Cit)$/CA system. According to reactions R1−R3 shown in Table 2, the $HO_2$ radical is produced upon irradiation and will partition to the gas phase. Figure 12(a) shows the $RH$ dependence of $HO_2$ production, $P_{HO_2}$, from thin films in CWFT experiment. We observed that $P_{HO_2}$ increased with $RH$ when the $RH$ was increased from 13 % to 29 % by a factor of about 2. This may be expected since an increase from 13 % to 29 % $RH$ leads to increasing molecular diffusion coefficients and faster chemical cycling (Lienhard

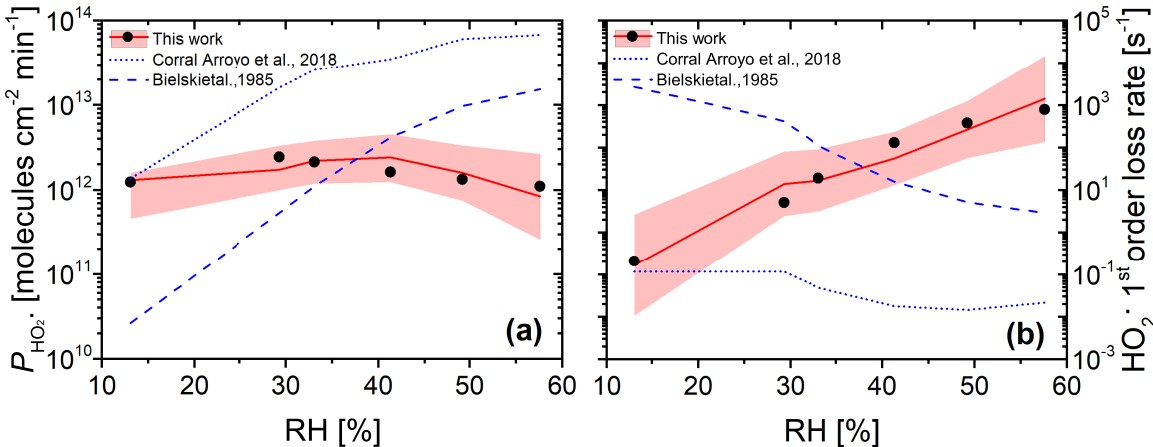

**Figure 12.** (a) Measured (black dots, experimental uncertainty of 3 % smaller than symbol size) and modelled (red line and shaded area) $HO_2$ production from continued photochemistry of $Fe^{III}(Cit)$/CA (molar ratio of 0.07) films as a function of $RH$ in CWFT experiments. Dotted and dashed blue lines were determined from parameterization of $k_5$ as a function of $RH$ (Corral Arroyo et al., 2018) and $k_5 = 8.3 \times 10^5$ $M^{-1} s^{-1}$ in dilute aqueous solution (Bielski et al., 1985), respectively. (b) The maximum $HO_2$ first order loss rate determined from the result of the maximum $HO_2$ concentration times $HO_2$ self reaction rate, $k_5$, which was adjusted to match observations in panel (a) (black dots), or times $k_5$ parameterized as a function of $RH$ for our PRAD model (Eq. (A11)) with confidence interval of 95 % (red line and shading), or times $k_5$ from Corral Arroyo et al. (2018) (dotted blue line), or $k_5 = 8.3 \times 10^5 \, M^{-1} s^{-1}$ (Bielski et al., 1985) (dashed blue line).

et al., 2014; Song et al., 2016). However, at $RH$ between 30 % and 60 %, $P_{HO_2}$ decreased with $RH$, with a production rate at 60 % similar to the one under dry conditions. This is probably due to the decreasing concentrations of donors (e.g., $Fe^{III}(Cit)$ and $Cit^{\cdot}$). But the decrease in concentrations with $RH$ is too small compared the increase in diffusion coefficients with $RH$.

This means that there must be a strong sink of $HO_2$ in the condensed phase when RH increases, which has been confirmed by an increasing $HO_2$ first order loss rate as shown with black dots in Figure 12(b). When predicting $P_{HO_2}$ with the PRAD model using a constant $HO_2$ self reaction rate of $k_5 = 8.5 \times 10^{-5} \, M^{-1} s^{-1}$ (Bielski et al., 1985) or a linearly increasing $P_{HO_2}$ with $RH$, as in Corral Arroyo et al. (2018), the model deviates significantly from the data and does not exhibit the observed trend of decreasing $P_{HO_2}$ for $RH > 30$ %. Both assumptions lead to a continuous increase $P_{HO_2}$ with $RH$ as indicated as

dashed and dotted blue lines in Fig. 12(a), with a decrease $HO_2$ first order loss rate as shown in Fig. 12(b), which is opposite from our adjustment. Therefore, we argue that the effects of the increasing diffusivity and the stronger sink of $HO_2$ with $RH$ compensate each other, making $P_{HO_2}$ almost independent of $RH$.

### 3.5 Photochemical degradation under atmospheric conditions

The PRAD model was developed to also be used in more general particle systems. After establishing a parameter set for the

475 PRAD model framework which satisfactorily explains the experimental data obtained with three complimentary experimental

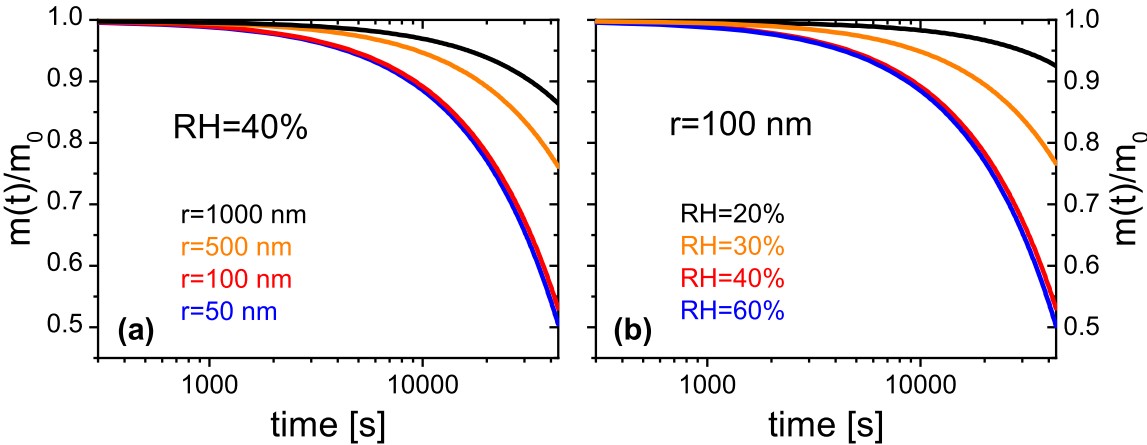

**Figure 13.** PRAD simulated organic mass loss of $Fe^{III}(Cit)$/CA (molar ratio of 0.01) for (a) particles of different radii at 40% RH, and (b) particles with a radius of 100 nm at different $RH$. All simulations for 293.5 K and 12 h solar irradiation at $30°$ zenith angle (Madronich et al., 1995).

techniques over a wide parameter range, we used the model for predicting photochemical degradation of organic aerosol particles containing carboxylate complexes. For example, an aqueous organic acid particle with a dust inclusion containing iron, may exhibit a low enough pH to dissolve part of the iron of the dust inclusion and form photo-reactive iron carboxylate complexes (George et al., 2012, 2015). If we take the PRAD model as being representative for such a class of particles, we may

estimate the degradation of the organic mass by volatilization of products to the gas phase. Figure 13 show examples of such PRAD model predictions: we assume that the organic mass of the particle is composed of aqueous $Fe^{III}(Cit)$/CA with a molar ratio of the latter being 0.01. Here panel (a) illustrates size dependence and panel (b) illustrates RH dependence for particles exposed to air at 293.5 K and an actinic flux at $30°$ zenith angle (Madronich et al., 1995). After 12 hours exposure to sunlight, the organic mass of the particle (here citric acid) has been photochemically processed to $CO_2$ and smaller compounds of high

volatility yielding a combined mass loss to the gas phase of up to 50 % depending on size and RH.

To better understand the importance of mass loss in this generalized system, we make a simple and modest comparison of mass loss in Fig. 13 on the order of 20% over 5 hours with ambient mass accumulation measured in the field. Our results are equal to a mass loss rate of about $0.4 \, \mu g \, m^{-3} \, (air) \, hr^{-1}$ assuming an aerosol population with an organic mass of $\sim 10$ $\mu g \, m^{-3} \, (air)$ undergoing iron-carboxylate photochemistry. This is much larger than observed organic mass accumulation in

ambient air masses due to photochemical aging during atmospheric transport at about $0.06 \, \mu g \, m^{-3} \, hr^{-1}$ or $6 \, \mu g \, m^{-3}$ over 4 days (Zaveri et al., 2012; Moffet et al., 2012). This implies, that the mass loss rates are fast enough to affect the balance between aerosol mass accumulation and loss. Hence, photochemical degradation may be very significant in iron containing organic aerosol, perhaps even more important than degradation through reactions with gas phase oxidants. Note, that our

model does not fully capture the acceleration of degradation as discussed in Sect. 3.1. Therefore, we argue that the degradation presented in Fig. 13 represents a lower limit of the expected degradation under atmospheric conditions. Clearly, panel (a) indicates that smaller particles degrade significantly faster than larger ones. However, the size dependence is more complex than a simple square law expected if reactions were purely limited by condensed phase diffusion. For particles with a radius larger than 50 nm, the time to re-partition 10 % of the mass to the gas phase depends almost linearly on size for these specific conditions. Panel (b) shows that photochemical processes are faster at higher RH corresponding to less viscous particles. The relative mass loss after 12 h irradiation between 20 % and 60 % RH is almost an order of magnitude larger for the highest RH compared to dry conditions, because of transport limitations at low RH. However, it should be noted that the relative mass loss at higher humidities (between 40 % and 60 % RH) is quite similar. Here, the photochemical degradation is barely limited by condensed phase diffusivity, but by iron availability in the particle. These simulations show the potential of photochemical degradation under atmospheric conditions, a systematic study exploring the whole range of atmospheric conditions is beyond the scope to this work.

## 4   Conclusions

We used three complimentary experimental techniques to characterize the impact of reduced mobility of aerosol constituents on photochemical degradation in highly viscous particles. As an atmospherically relevant model system, we chose aqueous $Fe^{III}(Cit)$/CA particles. These three experimental techniques investigated specific aspects of this photochemical reaction system. In EDB experiments, we measured the mass loss relating to the continual production and loss of $CO_2$ and other volatile products. We observed very significant condensed phase degradation and strong acceleration of the degradation rate with time. Further studies are needed to quantify all atmospheric implications, but our study suggests that photochemistry in iron containing organic aerosol will lead to a significant re-partitioning of condensed phase mass to the gas phase. We used STXM/NEXAFS to directly measure iron oxidation state in-situ with an environmental microreactor. These experiments yielded valuable information about where iron photochemical reduction and re-oxidation reactions took place, namely only very close to the surface, and it allowed to characterize to which degree iron compounds diffused inside single particles. We show that $O_2$ uptake and diffusion into a particle is a limiting factor considering the reactions required to produce species with an oxidative potential. In addition, we found that the direct $O_2$ reaction with iron(II)-organic complexes does occur and generate radicals inside the particle. Flow tube experiments performed on thin $Fe^{III}(Cit)$ films showed continuous production of $HO_2$, revealing a radical source inside the particles driven by photochemistry.

All data were used to constrain equilibrium and kinetic parameters as well as reaction rate coefficients in a new photochemical reaction and diffusion (PRAD) model with sufficient complexity to allow comparison with data of all experiments simultaneously. In particular, we were able to constrain the photolysis rate of $Fe^{III}$ due to the use of various light sources with various spectral intensities, while capturing the photochemical reduction. In addition, we determined the $HO_2$ production rate and its first order loss rate, and the diffusivity of $O_2$ in aqueous $Fe^{III}(Cit)$/CA system as a function of $RH$ and $Fe^{III}(Cit)$/CA molar ratio with a choice of $O_2$ related reaction rate coefficient and $O_2$ Henry's law coefficient.

Although a systematic study exploring the whole range of atmospheric conditions was beyond the scope to this work, there are some aspects of the PRAD model and certain parameters that we argue are reliable and pertainate to atmopsheric aerosol photochemistry. First, coefficients in the PRAD model framework can be changed to predict mass loss rates of a different iron-carboxylate complex system. We are fairly confident that diffusion coefficients of $CO_2$ and $H_2O$ can be used for atmospheric aerosol particles as these were obtained in a more targeted study (Dou et al., 2019). Mass loss rates in general are fairly reliable to be used in atmospheric particles as these are linked to photochemical reaction rates which have been characterized (Weller et al., 2013, 2014). Finally, reoxidation rates and production of radicals are also reliable, as the system is largely reacto-diffusion limited (see appendix A4) and these rates occur on the same scales as observed mass loss rates. In our companion paper (Alpert et al., 2020, under review), we show a detailed analysis of radical concentrations in ambient aerosol particles for a range of atmospheric conditions and iron content. However, our model still needs major improvements, such as including peroxyl radical chemistry and better constraints on individual parameters such as diffusion coefficients and reaction rate constants. The overall rate may be well-constrained by our experimental studies, however more targeted observations may be necessary for an accurate representation of $O_2$ chemistry, solubility and molecular transport independently of each other within aerosol particles.

The chemical evolution of the organic species resulting from the continual photochemical oxidation was not the scope of the present study. A separate study focusing on individual OVOCs and condensed phase products is currently underway that will allow to better constrain the chemical regimes and the evolution of the oxidation state of the organic fraction with time. This will also allow to assess more details of organic peroxy radical chemistry that help to explain the observed 'missing' $HO_2$ sink in this system. Furthermore, testing the PRAD model with different organic carboxylate ligands is desirable to broaden its applicability.

Using the PRAD model for predicting photochemical degradation for iron containing organic aerosol under atmospheric conditions let us conclude that this pathway of re-partitioning condensed phase mass to the gas phase is important and its regional and global impact should be investigated in further modelling studies. The PRAD model may serve as a basic framework for the chemistry and transport of compounds in single particles for such studies.

*Code and data availability.* The data that support the findings of this study are available from the corresponding authors upon request. The PRAD model code are publicly available and accessible here: https://doi.org/10.3929/ethz-b-000451609, and from the corresponding author upon request.

*Video supplement.* The video supplement related to this article is available online at: https://doi.org/10.5446/47955.

## Appendix A

### A1 Parameterization of $D_1^j(x, T, a_{aw})$

The liquid phase diffusion coefficients, $D_1^j(x, T, a_w)$, where $j$ is an index for all species, depend on $RH$, $T$ and the molar ratio, $x$, between $Fe^{III}(Cit)$ and CA. $D_1^j(x, T, a_w)$ was scaled with the diffusion coefficient of water in $Fe^{III}(Cit)$/CA aqueous system, $D_1^{H_2O}(x, T, a_w)$ using a scaling factor, $l_f^j$, following

$$l_f^j = \frac{D_1^j(x, T, a_w)}{D_1^{H_2O}(x, T, a_w)}. \tag{A1}$$

$D_1^j(x, T, a_w)$ has not been previously determined for aqueous $Fe^{III}(Cit)$/CA although, solution viscosity has and was found to be higher than aqueous CA solutions when $x > 0.05$ (Alpert et al., 2020, under review), implying slower molecular transport. Lienhard et al. (2014) reported the diffusion coefficient of water in aqueous CA without iron (i.e. $x = 0$), $D_{CA(aq)}^{H_2O}(T, a_w)$. In order to determine $D_1^j(x, T, a_w)$ in the PRAD model for a single experiment with a fixed value of $x$, $D_1^{H_2O}$ was scaled with $D_{CA(aq)}^{H_2O}(T, a_w)$ using another factor, $f_s$, following

$$D_1^{H_2O}(x, T, a_w) = D_{CA(aq)}^{H_2O}(T, a_w)f_s(x), \tag{A2}$$

where

$$\log f_s(x) = -0.7106 e^{-\frac{1}{4x}}. \tag{A3}$$

The diffusion coefficient of $CO_2$ at $T = 20\,°C$ and $x = 0.05$, $D_\circ^{CO_2}$, has been independently measured by Dou et al. (2019) as a function of $a_w$. We parameterized $D_\circ^{CO_2}$ as

$$D_\circ^{CO_2} = \left(D_\circ^{CO_2}(a_w = 1)\right)^{\alpha \cdot a_w} \cdot \left(D_\circ^{CO_2}(a_w = 0)\right)^{1 - \alpha \cdot a_w}, \tag{A4}$$

where $D_\circ^{CO_2}(a_w = 0) = 1.19 \times 10^{-16}\,m^2\,s^{-1}$ is the diffusion coefficient of $CO_2$ at $T = 20\,°C$, $x = 0.05$ and $a_w = 0$. The $T$ dependent diffusion coefficient of $CO_2$ in water in Eq. (A4) is

$$D_\circ^{CO_2}(a_w = 1) = D_0\left(\frac{T}{T_s} - 1\right)^m, \tag{A5}$$

where $D_0 = 1.39 \times 10^{-8}\,m^2\,s^{-1}$, $T_s = 227.0\,K$ and $m = 1.7094$. In the exponent terms of Eq. (A4),

$$\alpha = e^{(1 - a_w)^2(A + B \cdot a_w)}, \tag{A6}$$

where $A = 0.2824$, and $B = -1.8086$. In order to introduce a $x$ dependence of $D_1^{CO_2}$ for various experiments reported here, $f_s$ from Eq. (A3) for water was applied, following

$$D_1^{CO_2}(x, T, a_w) = D_\circ^{CO_2}(T, a_w)f_s(x). \tag{A7}$$

For all other species (excluding $H_2O$ and $CO_2$), $D_1^j(x, T, a_w)$ was determined using Eq. (A1) with

$$\log l_f^j = -0.7710 M_j^{\frac{1}{3}} - 1.4732, \tag{A8}$$

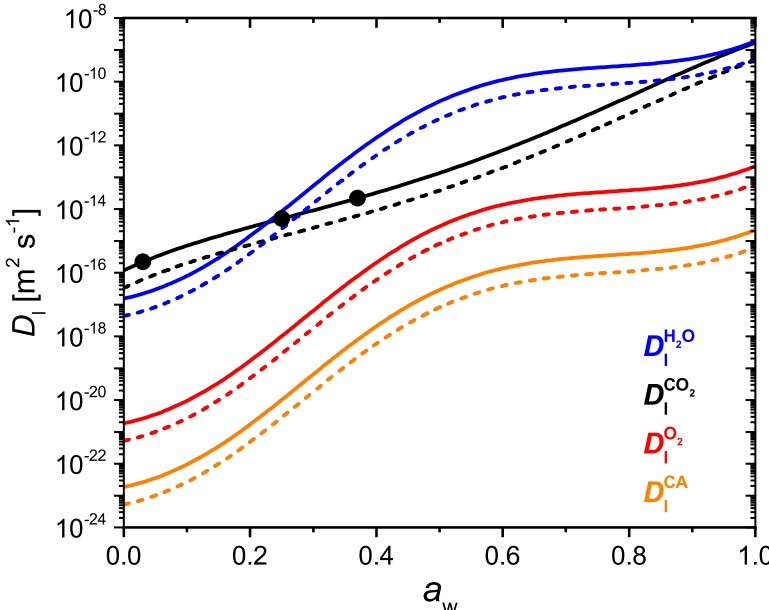

**Figure A1.** Parameterized liquid phase diffusion coefficients of $H_2O$ (blue), $CO_2$ (black), $O_2$ (red), and CA (orange) as a function of water activity at $Fe^{III}(Cit)/CA$ molar ratio of 0.05 (solid lines) and 1.0 (dashed lines). Three black dots are measured $CO_2$ diffusivity in $Fe^{III}(Cit)/CA$ particle with a molar ratio of 0.05 (Dou et al., 2019). The temperature is all at 293.5 K.

where, $M_j$, is molar mass. Diffusion coefficients of $H_2O$, $CO_2$, $O_2$, and CA as a function of $a_w$ at $x = 0.05$ and 1.0 are plotted in Fig. A1. Generally, $D_l^j(x, T, a_w)$ always decreases when $RH$ or $T$ is lowered. $D_l^j(x, T, a_w)$ decreases when $x$ increases beyond 0.05, but remains relatively constant otherwise. One caveat to using Eqs. (A1)−(A8) to calculate $D_l^j$ is that mass loss will ultimately lead to an increase in $x$, however, the PRAD model keeps $D_l^j$ fixed throughout the course of a model run. Since we observed increasing mass loss rates over time, any decrease in $D_l^j$ leading to slower chemical cycling due to increasing $x$ was likely a minor effect. Moreover, the product distribution and any effect on diffusion coefficients was unknown, so further time-resolved adjustments to $D_l^j$ were not considered. We suggest future studies investigate how molecular transport changes over the photochemical lifetime of iron-carboxylate complexes.

## A2  Parameterization of $D_g^j$

The gas phase diffusivity of each species $j$, $D_g^j$, was approximated via its molar mass ($M_j$) compared to that of water ($M_{H_2O}$),

$$D_g^j = D_g^{H_2O} \sqrt{\frac{M_{H_2O}}{M_j}}, \tag{A9}$$

with

$$D_{\mathrm{g}}^{\mathrm{H_2O}} = 0.211\left(\frac{T}{T_0}\right)^{1.94}\left(\frac{p_0}{p}\right), \tag{A10}$$

where $T_0 = 273.15$ K, $p_0 = 1013.25$ mb, and $D_{\mathrm{g}}^{\mathrm{H_2O}}$ is in $\mathrm{cm^2\,s^{-1}}$ (Pruppacher and Klett, 2010).

### A3 Parameterization of $k_5$

Based on the measurement of $HO_2$ production as a function of $RH$ using CWFT experiments, the $HO_2$ self reaction rate (R4 in Table 2), $k_5$, was adjusted so the PRAD model would exactly reproduce the data. We parameterized $k_5$ as a third degree polynomial function of $RH$ (%):

$$\log k_5 = -2.854 \times 10^{-5}RH^3 + 0.0024RH^2 + 0.1087RH - 0.05018, \tag{A11}$$

as shown in Fig. A2.

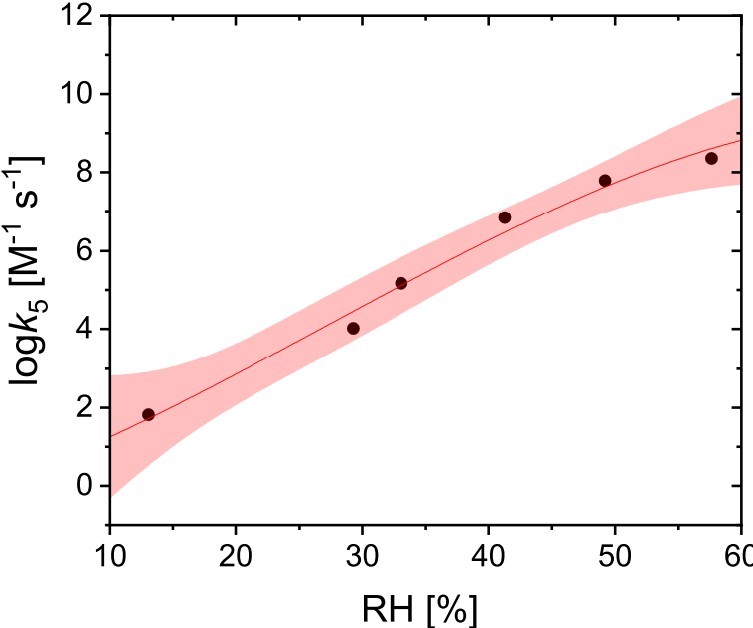

**Figure A2.** Adjusted values of $HO_2$ self reaction rate used in the PRAD model to exactly reproduce measured $P_{\mathrm{HO_2}}$ for CWFT experiments are shown as black dots. The red line and shading is the new parameterization (Eq. (A11)) and confidence intervals at 95 % for the $HO_2$ self reaction rate as a function of $RH$.

## A4  $\beta$ estimated by a reacto-diffusive framework

It is well known that multi-phase reactions can follow a reacto-diffusive kinetic regime (Alpert et al., 2019; Steimer et al.,
2014). For reacto-diffusive limitations as observed in Fig. 11, the $Fe^{III}$ fraction as a function of time was calculated as

$$\beta(t) = 1 - (1 - Kt)^2, \tag{A12}$$

with the assumption that $\beta_0 = 0$, i.e. there was no $Fe^{III}$ at $t = 0$.

$$K = \frac{3(H_{O_2}\sqrt{D_l^{O_2}})p_{O_2}\sqrt{k_{O_2}}}{\sqrt{[Fe_{tot}]}d_p}, \tag{A13}$$

where $H_{O_2}$ and $D_l^{O_2}$ are Henry's law coefficient and diffusion coefficient for $O_2$ in the $Fe^{III}(Cit)$/CA matrix, $p_{O_2}$ is the
pressure of $O_2$, $k_{O_2}$ is the total reaction rate of $O_2$, and $d_p$ is diameter of the single particle. $\beta = \frac{1}{e}$ when

$$t_{\frac{1}{e}} = \frac{1 - \sqrt{1 - \frac{1}{e}}}{K} = \frac{\left(1 - \sqrt{1 - \frac{1}{e}}\right)\sqrt{[Fe_{tot}]}d_p}{3(H_{O_2}\sqrt{D_l^{O_2}})p_{O_2}\sqrt{k_{O_2}}}. \tag{A14}$$

In a typical EDB experiment, $d_p = 20$ μm, $H_{O_2} = 3.5 \times 10^{-2}$ M atm$^{-1}$, $p_{O_2} = 8 \times 10^4$ Pa $= 0.789$ atm, and $k_{O_2} = 0.05$
M$^{-1}$ s$^{-1}$. $[Fe_{tot}]$ is 0.3192, 0.2763, and 0.2345 M at 24 %, 48 %, and 65 % RH, respectively. From EDB data points in Fig. 11,
we estimated that $\frac{1}{e}$ of $Fe^{III}$ can be fully recovered after around 1.7 h at 48 % RH and 0.45 h at 65 % RH. Thus the diffusion
coefficient of $O_2$ can be estimated to be $3.6 \times 10^{-16}$ and $4.4 \times 10^{-15}$ m$^2$ s$^{-1}$ at 48 % and 65 % RH, respectively, which are both
one order of magnitude less than the values from PRAD model prediction, but still consistent with each other when considering
all uncertainties. It should be noted that in Eq. (A14), actually $H_{O_2}\sqrt{D_l^{O_2}k_{O_2}}$ is the constraint, thus any uncertainty in $H_{O_2}$
or $k_{O_2}$ can change $D_l^{O_2}$.

## A5  Sensitivity of the PRAD model to various model parameters

As discussed in section 2.5 we performed manual tuning of the PRAD model parameters to reach satisfactory agreement with
all experimental data simultaneously. To show the sensitivity of the PRAD model results to a few of its parameters Fig. A3
shows again the data of the photocatalytic degradation experiment at 46 % RH described in section 3.1. In addition we show the
output of the model as well as model outputs obtained by varying one of the parameters by the indicated factors while keeping
all other parameters constant. Clearly, the sensitivities of the model output to varying these parameters are very different: while
the model output is quite sensitive to varying the rate constant of for the oxidation of $Fe^{II}(HCit)$ (R9 in Table 2) as well as to
oxygen diffusivity, the sensitivity to the equilibrium constant E8: $Fe^{2+} + HCit^{2-} \rightleftharpoons Fe^{II}(HCit)$ is significantly smaller and
the model is basically insensitive to the equilibrium constant E5: $Fe^{3+} + Cit^{3-} \rightleftharpoons Fe^{III}(Cit)$. In other words, this experiment
alone allows to constrain R9 or oxygen diffusivity as long as the other parameter is known, but does not allow constraining the
constants for the two equilibria.

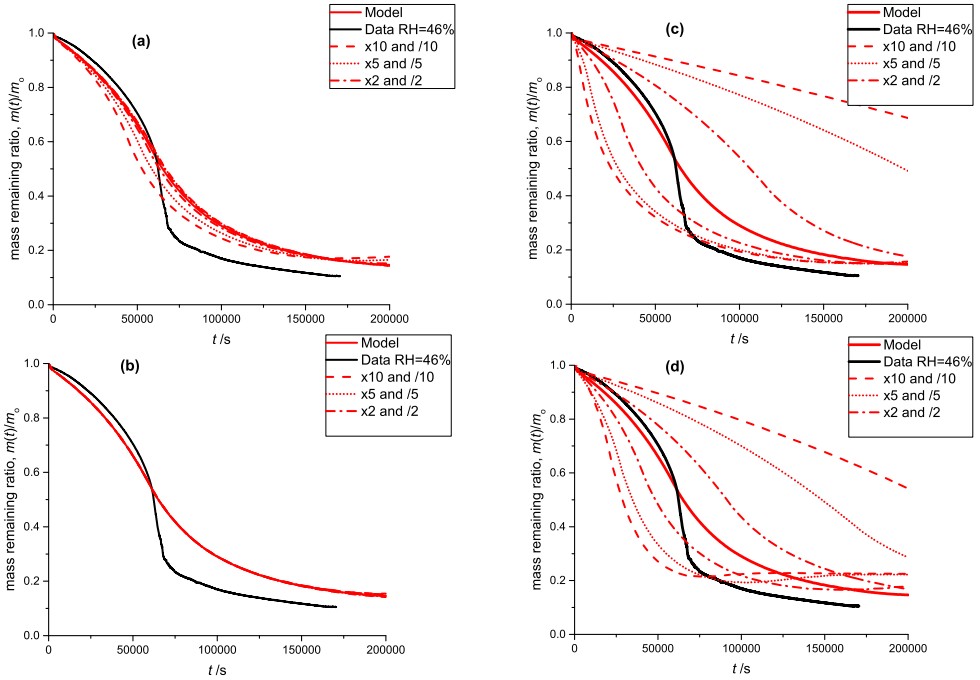

**Figure A3.** Black: mass remaining data versus time of the photo-degradation experiment discussed in section 3.1; RH 46 %. Red lines: model outputs obtained by varying a single parameter with factors: 10 and 0.1 (dashed), 5 and 0.2 (dotted), 2 and 0.5 (dash-dotted), 1 (solid). In panel (a) equilibrium constant of E8 is varied, in panel (b) equilibrium constant of E5 is varied, panel (c) shows the sensitivity to reaction constant R9 (see Table 2) and panel (d) the sensitivity to oxygen diffusivity (Table 1).

## A6   The full EDB data set showing the $RH$ effect on photocatalytic degradation efficiency

Figure 6 in the main text shows the photochemical degradation up to a mass remaining ratio of 0.6 and compares its temporal evolution with the PRAD model output. Here, we show in Fig. A4 for completeness the full experimental data set. Clearly seen is the shift to longer time scales with decreasing RH and that the PRAD model is no longer capable of capturing accurately the mass loss once about 20% of the initial mass is lost. However, we may conclude that even for mass remaining ratios lower than 0.5, the experimental data show a very significant shift of the degradation time scales with RH, but the total photochemical mass loos remains of the same order of magnitude, independent of RH.

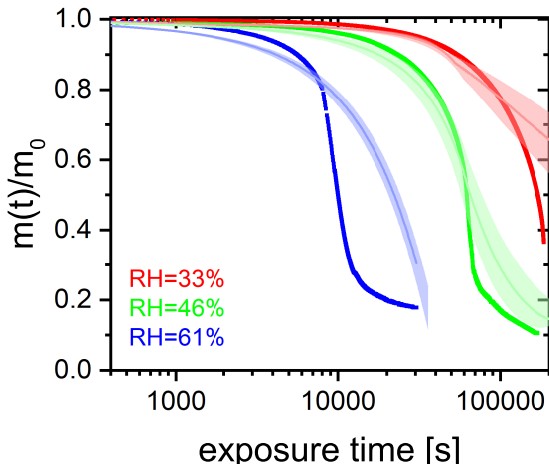

**Figure A4.** $Fe^{III}(Cit)/CA$ (molar ratio 0.05) particle mass change with irradiation time (log scale) at different RH: 33 % (red), 46 % (green), and 61 % (blue). The irradiation wavelength was 375 nm, its intensity was $0.25\ W\,cm^{-2}$, and the experimental temperature was 293.5 K. Thick lines are EDB experimental data. Thin lines are PRAD model outputs (with $\pm2$ % RH uncertainty shown as shaded area).

*Author contributions.* J.D. wrote the manuscript. M.A. and U.K.K. conceptualized and planned the study. J.D. conducted EDB experiments, water activity measurements, and data analysis and interpretation supervised by U.K.K. P.A.A. planned and conducted STXM/NEXAFS experiments supervised by B.W. and M.A. J.D., P.C.A., J.X., T.H., C.N.B. and K.D.H. also conducted STXM/NEXAFS experiments. B.W., J.R. and P.A.A. conducted STXM/NEXAFS data analysis and interpretation. P.C.A conducted CWFT experiments and data analysis and interpretation supervised by M.A. F.S. conducted viscosity experiments and data analysis and interpretation supervised by P.A.A. B.L. wrote and developed the PRAD model with the assistance of J.D. and P.A.A. H.H. advised on the photochemical reaction mechanisms. All co-authors discussed the results and commented on the manuscript.

*Competing interests.* The authors declare that they have no conflict of interest.

*Acknowledgements.* We thank Nir Bluvshtein for helpful discussions. We acknowledge funding by the Swiss National Science Foundation (grant no. 200021 163074/1). The PolLux end station was financed by the German Ministerium für Bildung und Forschung (BMBF) through contracts 05K16WED and 05K19WE2.

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
