# Peer review of "Photochemical degradation of iron(III)-citrate/citric acid aerosol quantified with the combination of three complementary experimental techniques and a kinetic process model"

_Atmospheric Chemistry and Physics, 2020_

## Referee Comment (RC1) · Anonymous Referee #1 · 12 Aug 2020

The manuscript seeks to explore the effect of high viscosity on photochemical processes using iron(III)-citrate (FeIII(Cit)) as a model light absorbing iron carboxylate complex. To investigate kinetic transport limitations, three complementary experimental approaches were used: (a) The mass loss of single, levitated particles was measured with an electrodynamic balance, (b) the oxidation state of deposited particles was measured with X-ray spectromicroscopy, and (c) HO2 radical production and release into the gas phase was observed in coated wall flow tube experiments. Also, a numerical multi-layered photochemical reaction and diffusion (PRAD) model that treats

chemical reactions and transport of various species was developed.

Overall, the experimental and modeling approach is novel and the study is publishable after addressing the following major and minor comments:

Major comments:

Line 126-127: Figure 2 shows the data while irradiation, not while the particle is equilibrating. The data for this part of the experiment are shown over a much shorter time than the 10 h stated in the methods section. what is the magnitude of mass loss during the dark equilibration period?

There are conclusions made in the abstract that did not come across very clearly upon examining the figures and the text. It is important that each major statement on observations in the manuscript is supported by the data analysis in the manuscript. The figures need to be re-made to make conclusions easier to see for connection with the abstract. For example, the conclusion in line 17 was for 10 micron particles over 24 hr, but the modeling results in Figure 13 were for 1 micron to nanometer particles, and their mass loss rate is different. Also, data is Figure 6 show less than 80% mass loss.

In the abstract, the statement, "The PRAD model was tuned to reproduce all experimental results": The mass loss data in Figure 2 does not show modeling results specific to the experimental setting to support this sentence. Also, model results in figures 7 and 11 are way off from the experimental data. So, it is not clear how this model reproduced the experimental data.

It is not clear how the molar ratio of Fe(III)(Cit) to citric acid (CA) was controlled. The manuscript shows that it was not fixed in the experiments, and the chemical composition was calculated. Section 2.1 does not describe solution preparation from which aqueous droplets were taken for the EDB measurements. The caption of the figures reported 0.01, 0.05, 0.07, and 1. How does these calculated molar ratios correlate with the solution prepared in the lab?

What is the pH of the solution prepared in the lab and that calculated from the chemical composition model? What is the effect of pH on mass loss? Photochemical degradation of iron complexes is pH dependent, not only wavelength dependent. This is a major issue that is not addressed in the current version of the manuscript.

There was no experimental studies looking at the extent of mass loss as a function of molar ratio, as there was for particle radius and %RH (e.g., figure 13).

Figure 10: per the mechanism in Figure 1, and the abstract line 4,5 "In the presence of $O_2$, ensuing radical chemistry leads to further decarboxylation, and the production of .OH, $HO_2$, peroxides, and oxygenated volatile organic compounds, contributing to particle mass loss", I expected the opposite result for the 'mass remaining' data in the right axis: that the mass remaining for the particle recovered in $O_2$ for 45 min is lower than irradiating fresh particle under $N_2$ since the photolysis and loss of $CO_2$ is driven by the chemistry of dissolved $O_2$. I did not find an explanation in the text for this part of the figure.

Minor:

Line 418: The reference Duo et al 2020 is not published. The sentence citing this reference does not refer to unknow information. There are book chapters and numerous review articles on this topic, which should be cited.

Figure 2: for consistency with the text, change the y-axis to hours instead of seconds for time since rates and time discussed in the text are expressed in hours.

Figure 3: there are two shaded areas in dark and light green, and only the light green is described in the legend. There seems to be a missing compound that describes the dark green shaded area. Also, it is not clear why there are references in the caption? Were some of the spectra shown adapted from these references?

Figure 6: the color description for RH does not match the actual colors of the markers. For example: in the figure, data for RH=33% is red, and in the caption is it mentioned

[Figure]

as '(black)'. In be, there are markers and lines, for experimental data and model results. Modify the legend to reflect that to avoid confusion which is which.

Figure 11: use different marker shapes for different RH, chemicals. Take into account that this figure might be printed on a black and white printer and will be hard to read as is. Why is here more than one data point at t $\sim$ 3 min for the RH = 48%? Same at t $\sim$ 15 min and 20 min for 48% and 65% for iron(II) and iron (III) citrate?

Figure 12: add a legend defining the markers and lines

Figure 13: add 'RH=40%' to legend in (a), and '100 nm' to legend in (b).

———————————————

---

## Referee Comment (RC2) · Anonymous Referee #2 · 14 Sep 2020

This manuscript (acp-2020-779) investigates iron(III) carboxylate photochemistry using three different experimental approaches. In an electrodynamic balance, the mass loss of a levitated droplet containing Fe(III)-citrate and citric acid upon irradiation was quantified. Using STXM/NEXAFS, changes to the oxidation state of deposited Fe-containing aerosol were measured. With a coated wall flow tube, the production of HO2ů was monitored during photochemical reaction through a chemiluminescence measurement of NO. The Photochemical Reaction and Diffusion (PRAD) model was then applied to all three sets of experimental data to explore chemical processes occurring within a

single droplet. The results indicate that photochemical degradation and repartitioning of molecules to the gas phase could be very significant for Fe-containing aerosols in the atmosphere.

This manuscript combines interesting and well-conducted complementary experiments with a new modelling approach to develop a consistent interpretation of the complex chemical processes occurring within an individual droplet, representing an important advance. The manuscript is well-written, with clearly constructed and appropriate figures. This manuscript is within the scope of Atmospheric Chemistry and Physics and will be publishable once the comments below are addressed.

Comments:

1. PRAD model: The PRAD model appears to be a novel model to describe chemistry within a single droplet. The manuscript would be improved by providing some additional context identifying the distinct aspects of the PRAD model relative to existing models (e.g. KM-GAP, Kinetiscope) which appear to operate under similar principles. In addition, the authors indicate that the model is manually tuned. The authors should clarify how they minimized the possibility of tuning to a local minimum. Moreover, in discussing the number of parameters tuned and the fraction of parameters that are poorly known, the authors should also provide some indication about the sensitivity of the model to these parameters. This is discussed a bit on pages 21 and 22, but additional detail would strengthen the manuscript.

2. Page 12, line 247: In this line, the number of molecules is defined as $N_n$, whereas on the previous page (line 221), the number of moles is defined as $N_i$. Although the discussion for each revolves around different portions of the particle (n representing the outermost layer; i representing inner layers), the authors should ensure that terms are consistently used throughout the manuscript. Moreover, on page 12, line 270, $n*Cit$, $nCit$ and $nFeCit$ do not appear to be defined. Presumably these refer to molarity, in which case should not the appropriate term be c (as defined on page 9, line 205)?

3. Figure 6 caption: The colours listed in the caption do not match those in the figure.

4. Page 17, line 329 ("within about 6 hours"): This is not shown experimentally in Fig. 8. Experimental data at 60% RH only extend to ~3 h. The model extends to 6 h, but this is not an observation.

---

## Referee Comment (RC3) · Anonymous Referee #3 · 16 Sep 2020

The authors of manuscript acp-2020-799 studied the photochemical degradation of iron(III) carboxylates (with iron(III) citrate as a proxy) via 3 complementary methods. The mass loss of single particles was measured using an electrodynamic balance, the oxidation state of single particles was determined using STXM/NEXAFS, and $HO_2$ production was monitored using a coated wall flow tube experiment. These results were all used to validate a novel Photochemical Reaction and Diffusion (PRAD) model. The model reproduces experimental results within stated levels of uncertainty. The experimental data shows that O2 uptake and mobility, influenced by particle viscosity

and size, is a significant factor in the degradation of iron containing aerosols.

The combination of experimental and modelling work done together is a compelling way to debut the PRAD model. Manuscript acp-2020-799 is constructed well and supported with clear figures (both in the manuscript and SI) and should be published after the following comments are addressed.

Major Comments:

———————————————————–

1) The PRAD model was developed (presumably) for modeling iron degradation but the framework can be used in other particle/gas reaction systems. Some discussion on how general this model is/can be and which systems it could be successfully applied to would be good. It would also be helpful to add statements about what this model does well, like which parameters are known with the most certainty or which conclusions are the strongest, to help the reader understand the PRAD model's place among other similar models (both those already existing and those yet to be developed).

2) Some additional discussion about the sensitivity that the manual tuning of certain parameters in the PRAD model has would be useful. Quantifying the sensitivity would be excellent but at least something to help the reader gauge the effect that a slight mistuning might have. In a similar vein, adding a column to table 1 showing the uncertainties in each of these parameters (if they're available) would be good.

3) Line 322-323 and Figure 7 shows a statistical agreement with 95% confidence between the model and the data, but it is just on the edge of significance. Even though a 95% confidence level is commonly used, it is ultimately arbitrary. A sentence or two giving some extra context or how the estimate could improve would be helpful in making the conclusion (that jcalc is a good estimator for jobs) more robust.

Minor Comments:

———————————————————

[Figure]

Line 45: "humic like" should be hyphenated Line 57: "time resolved" should be hyphenated

Line 72: The chemical formula of iron(III) citrate is $FeC_6H_5O_7$ but the structural formula written shows an incorrect number of Carbon and Hydrogen atoms. Please revise the structural formula.

Line 99: reword: "The PRAD model allows to simulate"

Line 201: The denominator in the expanded version of equation (4) uses "r" as a subscript, it shows "$0.5(r_{r+1} - r_{i-1})$" should it read "$0.5(r_{i+1} - r_{i-1})$" instead? Also why use $0.5(r_{i+1} - r_{i-1})$ instead of $(r_{i+1} - r_i)$?

Line 209: Shouldn't the radius should be squared in equation (6) as part of the sphere surface area equation? This continues through equations (8), (9), (13), (14), and (16). A brief clarification would help.

Line 248: If $c_n$ is molar concentration (as defined in line 205), then $N_n$ should be the number of moles, not the number of molecules.

Line 270: Because concentration was defined as a subscripted "c" (i.e. $c_n$ in line 205) replace the nCit values with cCit for consistency.

Figure 6: The caption colors do not correspond to the figure colors.

Line 304: "The degradation progresses..." should be "The degradation processes..."

---

## Referee Comment (RC4) · Anonymous Referee #4 · 16 Sep 2020

The manuscript at hand describes a model and several sets of laboratory experiments that examine Fe driven photocatalytic organic chemistry in model particles. I am impressed with both the laboratory and modeling techniques used here. My main critique involves how applicable this system is issues of a global scale, as the paper attempts to address in the conclusions. How important are organic-Fe particles globally? Are they so abundant that they are expected to measurably affect the organic aerosol burden? How representative is the Fe citrate system of globally distributed aerosol iron? I comment below regarding the phase of Fe and how that might affect the conclusions

at the end of the paper. I think the paper is a good one because it does stimulate one to ask many questions. Moreover, I think there is potential to extend these methods beyond this particular system. My specific comments are as follows: Line 20: I suggest adding a brief description of the type of parameters used to "tune" the model. Line 119: It is stated that refractive index is assumed not to change. Shouldn't the refractive index change with the amount of water on the particle? If the RH changes, so will the refractive index. Was this accounted for? Refractive indices for both dry and aqueous phases should be available and used to show this is a good assumption. If its not a good assumption, how might it affect results and conclusions? Line 125: Molar absorptivities are stated, and will affect reaction rates. How likely is it for side products to be formed that have different optical properties? Line 126: I find the term "O2 gas phase" and "oxygen atmosphere" ambiguous and confusing. Are these atmospheres of pure O2? I think it would be best to spell this out. Line 190: Change "devides" to "divides" Line 329: Change "dryer" to "drier" Figure 13: How representative are these sizes of Fe containing particles? Also, if it is dust, how applicable are the results obtained on the iron citrate system which is homogeneously mixed and all soluble? Also, panel B is for a 200 nm diameter particle. Why not increase the particle size for a more reasonable Fe containing particle size? Line 425: How applicable are the EDM experiments to smaller particles? Is it possible to have a size dependent photolysis rate for reasons beyond that described in the PRAD model? What about small particle optical effects?

---

## Author Comment (AC1) · 17 Sep 2020

We thank the reviewer for such helpful comments and suggestions of our manuscript. We will answer the questions one by one below and list our planned changes to the manuscript as well.

Major comments:

 Line 126-127: Figure 2 shows the data while irradiation, not while the particle is equilibrating. The data for this part of the experiment are shown over a much shorter time than the 10 h stated in the methods section. what is the magnitude of mass loss during the dark equilibration period?

The mass of the particle is quite stable during the dark equilibration period as soon as the RH and T in the trap are stable, since no chemistry happening in the dark. **An example is shown below.** Therefore, we only show a short dark period to indicate that the particle was stable before we turned on the light.

No change to the manuscript.

[Figure]

There are conclusions made in the abstract that did not come across very clearly upon examining the figures and the text. It is important that each major statement on observations in the manuscript is supported by the data analysis in the manuscript. The figures need to be re-made to make conclusions easier to see for connection with the abstract. For example, the conclusion in line 17 was for 10 micron particles over 24 hr, but the modeling results in Figure 13 were for 1 micron to nanometer particles, and their mass loss rate is different. Also, data is Figure 6 show less than 80% mass loss.

We thank the reviewer for pointing out that the reader needs to be more clearly informed which results are experimental observations and which are model results.
Fig. 2 shows that a single, levitated particle with a 12 um radius loses about 80% of its mass to the gas-phase under the conditions of this particular experiment.
The purpose of Fig. 6 is to show that the PRAD model is able to simulate such experiments reasonably well. We show only the mass loss of the first 40 % here as eventually crystallization of iron citrate occurs within the particle, which is not treated in our model. However, we agree with the reviewer that it may help the reader to see the data of the subsequent mass loss to better understand our statement in lines 302-304:
*"However, the model is not able to capture the full degree of acceleration of the degradation rate, as it does not attempt to include the complete multi-generational oxidation chemistry at the level of individual components after initial radical production."*

The reviewer is correct, that the model results of Fig. 13 cannot be directly compared to the experimental data shown for example in Fig. 2 as the irradiation conditions and the particles size is quite different. For estimating the effect on atmospheric aerosol particles, we developed the PRAD model.

**Planned changes for a revised manuscript**:
To avoid confusion between experimental data and modeling predictions, we will shift lines 17 to 20 of the abstract to directly follow the list of experimental techniques and only then introduces the PRAD model in the abstract so that there is a clear separation between experimental results and modeling predictions.

The figure below shows how the experiments of Fig. 6 evolve subsequently. We will put this figure in the appendix of the paper. And a comparison with PRAD modeling is shown in the same figure as well.

[Figure]

Figure A3. $Fe^{III}$(Cit)/CA (molar ration 0.05) particle mass change with irradiation (log scale) at different RH: 33% (red), 46% (green), and 61% (blue). The irradiation wavelength was 375 nm, its intensity was 0.25 W cm$^{-2}$, and the experimental temperature was 293.5 K. Thick lines are EDB experimental data. Thin lines are PRAD model outputs (with ±2% RH uncertainty shown as shaded area).

In the abstract, the statement, "The PRAD model was tuned to reproduce all experimental results": The mass loss data in Figure2 does not show modeling results specific to the experimental setting to support this sentence. Also, model results in figures 7 and 11 are way off from the experimental data. So, it is not clear how this model reproduced the experimental data.

**Planned changes for a revised manuscript (changes in bold)**:
We will rephrase the sentences in the abstract and add a sentence: "The PRAD model was tuned to **simultaneously** reproduce all experimental results as closely as possible and captured the essential chemistry and transport during irradiation. In particular, the photolysis rate of $Fe^{III}$, the re-oxidation rate of $Fe^{II}$, HO$_2$ production, and the diffusivity of O$_2$ in aqueous $Fe^{III}$(Cit)/CA system as function of relative humidity and $Fe^{III}$(Cit)/CA molar ratio could be constrained. **This led to satisfactory agreement within model uncertainty for most, but not all experiments performed.**

It is not clear how the molar ratio of Fe(III)(Cit) to citric acid (CA) was controlled. The manuscript shows that it was not fixed in the experiments, and the chemical composition was calculated. Section 2.1 does not describe solution preparation from which aqueous droplets were taken for the EDB measurements. The caption of the figures reported 0.01, 0.05, 0.07, and 1. How does these calculated molar ratios correlate with the solution prepared in the lab?

We agree with the reviewer and wish to make it more clear which mole ratio was used for each experiment and each PRAD model run. To summarize, the initial mole ratio was 0.05 for all EDB experiments shown in the manuscript. For all STXM/NEXAFS experiments, the initial mole ratio was 1.0. For all CWFT experiments, the initial mole ratio was 0.07. The PRAD model of these experiments used the same initial mole ratio, respectively. We performed one additional PRAD model simulation (Fig. 13) using a mole ratio of 0.01. **We have added an addition section (new section 2.1) that details our solution preparation:**

Citric acid (≥ 99.5%) and Iron(III) citrate tribasic monohydrate (18–20% Fe basis) were purchased from Sigma-Aldrich. Iron(II) citrate ($Fe^{II}$(HCit)) was purchased from Dr. Paul Lohmann GmbH KG. Dilute aqueous solutions of $Fe^{III}$(Cit)/citric acid and $Fe^{II}$(HCit)/citric acid were made in ultrapure water (18 M Ω cm$^{-1}$, MilliQ). Since $Fe^{III}$(Cit) only dissolves slowly in water, citric acid solution with $Fe^{III}$(Cit) crystals inside has to be put in an ultrasonic bath for at least 24 hours, the same dissolving procedure was also applied to the $Fe^{II}$(HCit) powders. Note that all the procedures were done under red light illumination because $Fe^{III}$(Cit) is light sensitive. Molar ratio between $Fe^{III}$(Cit) and CA was different for each experimental methods used in this study. For EDB, STXM/NEXAFS and CWFT experiments, stock solutions were prepared with molar ratios of 0.05, 1.0 and 0.07, respectively.

What is the pH of the solution prepared in the lab and that calculated from the chemical composition model? What is the effect of pH on mass loss? Photochemical degradation of iron complexes is pH dependent, not only wavelength dependent. This is a major issue that is not addressed in the current version of the manuscript.

We thank the reviewer for pointing out and agree strongly with her/him about the importance of pH.

For all three kinds of experiments, CA is highly concentrated in the aqueous system at the relative humidities considered. Its presence dominates the pH (to values in the range of $1-2$), i.e. the conditions considered in the paper are always acidic. We checked on pH changes predicted by the PRAD model, as it treats the relevant equilibria. According to model outputs, the pH value changes less than 5% during the whole photochemical processes under constant RH.

**Planned changes for a revised manuscript (changes in bold)**:
We will add this clarification to the section of the PRAD model.

There was no experimental studies looking at the extent of mass loss as a function of molar ratio, as there was for particle radius and %RH (e.g., figure 13).

We did the experiments to look at the effect of molar ratio (thick lines), and compare the data with PRAD model (thin lines). However, the data was not taken into account when we were optimizing the model.

[Figure]

In addition, CWFT experiments were performed as a function of mole ratio and are included in our another manuscript, showing an increasing $HO_2$ production with the molar ration of $Fe^{III}(Cit)$ to CA, which is captured by PRAD model (Alpert et al., 2020).
*Alpert, P. A., Dou, J., Corral Arroyo, P., Schneider, F., Xto, J., Luo, B., Peter, T., Huthwelker, T., Borca, C. N., Henzler, K. D., Herrmann, H., Raabe, J., Watts, B., Krieger, U. K., and Ammann, M.: Anoxic aerosol particles leads to preserved radicals, under review of Nature Communication, 2020.*

Figure 10: per the mechanism in Figure 1, and the abstract line 4,5 "In the presence of O2, ensuing radical chemistry leads to further decarboxylation, and the production of .OH, HO.2, peroxides, and oxygenated volatile organic compounds, contributing to particle mass loss", I expected the opposite result for the 'mass remaining' data in the right axis: that the mass remaining for the particle recovered in O2 for 45 min is lower than irradiating fresh particle under N2 since the photolysis and loss of CO2 is driven by the chemistry of dissolved O2. I did not find an explanation in the text for this part of the figure.

We think the reviewer misunderstood Fig. 10. Fig. 9 is intended to illustrate the experimental procedure. We agree with the reviewer that the uptake of $O_2$ leads to further oxidation and thus more mass loss. However, both data sets (black and red) shown in Fig. 10 are mass loss due to photochemistry in $N_2$. The black one is for the irradiation of fresh particle, and the red one is for another irradiation of the particle after recovery in $O_2$ in the dark for 45 min, which shows less mass loss due to less photochemically reactive Fe(III) contained in the particle than it initially had.

**Planned changes for a revised manuscript (changes in bold)**:
We need to stress stronger that the data shown in Fig. 10 are due to loss in a nitrogen atmosphere where only the decarboxylation is occurring with no further chemistry. We will rephrase the corresponding section.

Minor comments:

Line 418: The reference Duo et al 2020 is not published. The sentence citing this reference does not refer to unknow information. There are book chapters and numerous review articles on this topic, which should be cited.

We thank the reviewer for the suggestion, and change the reference to:

*George, C., D'Anna, B., Herrmann, H., Weller, C., Vaida, V., Donaldson, D. J., Bartels-Rausch, T., and Ammann, M.: Emerg-605ing areas in atmospheric photochemistry, in: Topics in Current Chemistry, vol. 339, pp. 1–53, Springer, Berlin, Heidelberg, https://doi.org/10.1007/128_2012_393, 2012.*

*George, C., Ammann, M., D'Anna, B., Donaldson, D. J., and Nizkorodov, S. A.: Heterogeneous photochemistry in the atmosphere, Chem. Rev., 115, 4218–4258, https://doi.org/10.1021/cr500648z, 2015.*

Figure 2: for consistency with the text, change the y-axis to hours instead of seconds for time since rates and time discussed in the text are expressed in hours.

In the text, we discussed in both seconds and hours.
We change the upper x-axis to hours:

[Figure]

Figure 3: there are two shaded areas in dark and light green, and only the light green is described in the legend. There seems to be a missing compound that describes the dark green shaded area. Also, it is not clear why there are references in the caption? Were some of the spectra shown adapted from these references?

We will change the caption in Fig. 3 to be easier to understand and referenced correctly. First, we clarify that there is only green shading and blue shading. We do not have different shades of green shading.

**We will change the caption to be:**
Figure 3. Iron L-edge NEXAFS spectra of $Fe^{III}(Cit)/CA$ particles before and after irradiation with UV light shown as orange and red, respectively. The previously recorded spectrum from mixed xanthan gum (XG) and $FeCl_2$ particles exposed to ozone is shown as the purple line, and a spectrum from $FeCl_2$ particles is shown as the blue line (Alpert et al., 2019). $FeCl_2$ and $FeCl_3$ spectra from Moffet et al. (2012) are shown as the blue and green shading, respectively. The vertical dashed lines indicate peak X-ray absorption at 707.9 shifted to 708.3 eV for $Fe^{II}$ and 709.6 eV shifted to 710.0 eV for $Fe^{III}$.
.

Figure 6: the color description for RH does not match the actual colors of the markers. For example: in the figure, data for RH=33% is red, and in the caption is it mentioned as'(black)'. In be, there are markers and lines, for experimental data and model results. Modify the legend to reflect that to avoid confusion which is which.

We thank the reviewer for pointing out this mistake. The color descriptions in the caption have been corrected.

Figure 11: use different marker shapes for different RH, chemicals. Take into account that this figure might be printed on a black and white printer and will be hard to read as is. Why is here more than one data point at t~3 min for the RH = 48%? Same at t~ 15 min and 20 min for 48% and 65% for iron(II) and iron (III) citrate?

We use different marker shapes but also keep different colors to indicate model simulations under corresponding conditions.

[Figure]

The reason for more than one data point is that we repeated the same procedure several times to be confident of the experimental results.

Figure 12: add a legend defining the markers and lines

Adapted accordingly.

[Figure]

Figure 13: add 'RH=40%' to legend in (a), and '100 nm' to legend in (b).

Adapted accordingly.

---

## Author Comment (AC2) · 3 Nov 2020

**1 Reviewer 2 comments and replies**

1. PRAD model: The PRAD model appears to be a novel model to describe chemistry within a single droplet. The manuscript would be improved by providing some additional context identifying the distinct aspects of the PRAD model relative to existing models (e.g. KM-GAP, Kinetiscope) which appear to operate under similar principles. In addition, the authors indicate that

- 5 the model is manually tuned. The authors should clarify how they minimized the possibility of tuning to a local minimum. Moreover, in discussing the number of parameters tuned and the fraction of parameters that are poorly known, the authors should also provide some indication about the sensitivity of the model to these parameters. This is discussed a bit on pages 21 and 22, but additional detail would strengthen the manuscript.
- The KM-GAP model (Shiraiwa et al., 2012) and the PRAD model rely on the same kinetic model framework for aerosol
  surface chemistry and gas-particle interactions (Pöschl et al., 2007). Numerically, the PRAD model uses a Euler forward step method, while KM-Gap solves coupled differential equations. Of course, the PRAD model presented here uses a chemistry scheme targeted for the particular model system of this work. In contrast, Kinetiscope does not integrate sets of coupled differential equations to predict the time history of a chemical system. Instead, it uses a general stochastic algorithm to propagate a reaction. The algorithm executes a random walk through event space, where an event is an individual reaction or diffusion
  step, rather than physical space, to generate a fully accurate time history of the system (Houle et al., 2015).
- We will add to the beginning of section 2.5: "Conceptually, the PRAD model relies on the kinetic model framework for aerosol surface chemistry and gas-particle interactions (Pöschl et al., 2007), similar as for example the KM-GAP model (Shi-raiwa et al., 2012). Numerically, the PRAD model uses a Euler forward step method as explained in detail below, while KM-Gap solves coupled differential equations. In passing, there are alternative approaches, for example Kinetiscope (Houle
- 20 et al., 2015) does not integrate sets of coupled differential equations to predict the time history of a chemical system. Instead, it uses a general stochastic algorithm to propagate a reaction."

We completely agree with the reviewer that our manual tuning cannot ensure finding the global minimum. To better explain our procedure we will add the following paragraph to the end of section 2.5:

- "We restricted our tuning of the parameters to reach satisfactory agreement with all experimental data simultaneously. The equilibrium constants and rate coefficients that were tuned are indicated in Table 2. These parameters were adjusted in a wide and acceptable range until a good representation of our data could be obtained. For example, the fraction of iron(III) in a photoactive complex (equilibrium E5 in Table 2) must have been high enough to reproduce STXM/NEXAFS observations that iron could be reduced to low levels as seen in Fig. 7. In comparison, E7 must have been much lower than E5 so that the amount of iron(III) in a non-photoactive complex was small compared to being in complex with citrate. As another example,
- 30 oxidation of  $Fe^{2+}$  (R6-R8 in Table 2) is fairly well-referenced, and therefore, we adjusted the rate of reaction R9 until the model reoxidation rates matched those observed. Tuning of individual bulk diffusion coefficients for all species was not attempted. Instead, we simplified the representation of diffusion coefficients using a parameterization as function of molar mass described in appendix A1. The 2 constants in Eqn (A8) and 2 constants in Eqn (A3) were tuned resulting in the absolute diffusion coefficients shown in Fig. A1. Henry's law coefficients for gasses were tuned, however purposefully set at values higher than
- 35 expected for pure water or highly dilute aqueous solution. This was inspired by previous studies regularly reporting solubility of e.g.  $O_2$  and  $CO_2$  higher in a variety organic liquids than water (Fogg, 1992; Battino et al., 1983)."

As discussed in the answers to question by the other reviewers, we have strengthened the statement that individual parameters which have been tuned should be used with care by adding the following text.

"It is important to note that the result of this tuning does not mean that we found the global minimum in the parameter space, see e.g. (Berkemeier et al., 2017). A thorough search for a global minimum for our model with 16 tuning parameters for chemistry, 4 tuning parameters (and our parameterization) for diffusion and 9 tuning parameters for solubility is computationally very expensive and beyond the scope of this paper. However, for our purpose here, namely modeling typical timescales of photochemical degradation of organic aerosol under atmospheric conditions (see Sec. 3.5) the PRAD model framework should allow sufficiently accurate predictions. In other words, we expect similar mass degredation in atmospheric particles due to

45 the fact that many other relevant iron-carboxylate compounds undergo LMCT similarly as to our model system. Additionally, if a system requires parameter values that significantly different that ours, the PRAD model framework itself may still be valid. Note, that careful evaluation is needed when picking a single parameter of the PRAD model for use in another context. Comparison of the refined model with our experimental data are shown in the next section."

**Figure 1.** Black: mass remaining data versus time of the photo-degradation experiment discussed in section 3.1; RH 46 %. Red lines: model outputs obtained by varying a single parameter with factors: 10 and 0.1 (dashed), 5 and 0.2 (dotted), 2 and 0.5 (dash-dotted), 1 (solid). In panel (a) equilibrium constant of E8 is varied, in panel (b) equilibrium constant of E5 is varied, panel (c) shows the sensitivity to reaction constant R10 (see Table 2) and panel (d) the sensitivity to oxygen diffusivity (Table 1).

To allow the reader gaining a better feeling for the sensitivity of the model to various parameters we will add an additional appendix showing model results for increasing and decreasing single parameters of the model by up to one order of magnitude:

As discussed in section 2.5 we performed manual tuning of the PRAD model parameters to reach satisfactory agreement with all experimental data simultaneously. To show the sensitivity of the PRAD model results to a few of its parameters Fig. 1 shows again the data of the photocatalytic degradation experiment at 46 % RH described in section 3.1. In addition we show the output of the model as well as model outputs obtained by varying one of the parameters by the indicated factors with keeping

all parameters constant. Clearly, the sensitivities of the model output to varying these parameters are very different: while the model output is quite sensitive to varying the rate constant of for the oxidation of FeII(HCit) (R10 in Table 2) as well as to oxygen diffusivity, the sensitivity to the equilibrium constant E8: Fe2+ + HCit2− ⇒ FeII(HCit) is significantly smaller and the model is basically insensitive to the equilibrium constant E5: Fe3+ + Cit3− ⇒ FeIII(Cit). In other words, this experiment alone allows to constrain R10 or oxygen diffusivity as long as the other parameter is known, but does not allow constraining

the constants for both equilibria.

55

2. Page 12, line 247: In this line, the number of molecules is defined as Nn, whereas on the previous page (line 221), the number of moles is defined as Ni. Although the discussion for each revolves around different portions of the particle (n representing the outermost layer; i representing inner layers), the authors should ensure that terms are consistently used

65 throughout the manuscript. Moreover, on page 12, line 270, n\*Cit, nCit and nFeCit do not appear to be defined. Presumably these refer to molarity, in which case should not the appropriate term be c (as defined on page 9, line 205)?

We thank the reviewer for pointing this out. We will define Nn as the moles of molecules in the outermost shell to ensure consistence. We will change 'n\*Cit, nCit and nFeCit' to 'N\*Cit, NCit and NFeCit' to represent the moles of each species.

*3. Figure 6 caption: The colours listed in the caption do not match those in the figure.*

70 We thank the reviewer for pointing out this mistake. The color descriptions in the caption have been corrected.

4. Page 17, line 329 ("within about 6 hours"): This is not shown experimentally in Fig. 8. Experimental data at 60 % RH only extend to  $\sim$ 3 h. The model extends to 6 h, but this is not an observation.

We will render this sentence to be more precise: "While particles were observed to re-oxidize to 70 % within 2 hours at 60 % RH and expected to be completely re-oxidized within about 6 hours according to PRAD model simulations, no significant

75 re-oxidation occurred on this timescale for the particles exposed to only 40 % RH."

**2 Reviewer 3 comments**

80

1) The PRAD model was developed (presumably) for modeling iron degradation but the framework can be used in other particle/gas reaction systems. Some discussion on how general this model is/can be and which systems it could be successfully applied to would be good. It would also be helpful to add statements about what this model does well, like which parameters are known with the most certainty or which conclusions are the strongest, to help the reader understand the PRAD model's place among other similar models (both those already existing and those yet to be developed).

We are very happy about the reviewer's suggestion that our model framework could be applied in other systems. We will provide a more detailed discussion of this aspect as indicated by the reviewers suggestion. We first note, we have already chosen a system of interest (an organic aerosol particle with a dust inclusion, (see section 3.5, first paragraph) that is more

- 85 general than  $\text{Fe}^{\text{III}}(\text{Cit})$ . In order to expand the applicability of this system, we make a simple comparison of mass loss for this system on the order of 20% over 5 hours with ambient mass accumulation measured in the field. As a modest estimate, the results in Fig. 13 of our manuscript indicate this is equal to a mass loss rate of about 0.4 µg m-3 (air) hr-1 for an aerosol population with an organic mass of ~ 10 µg m-3 (air) undergoing iron-carboxylate photochemistry. This is much larger than observed organic mass accumulation in ambient air masses due to photochemical aging during atmospheric transport at about
- 90  $0.06 \ \mu g \ m^{-3} \ (air) \ hr^{-1}$  or  $6 \ \mu g \ m^{-3} \ (air)$  over 4 days (Zaveri et al., 2012; Moffet et al., 2012). This implies, that the mass loss rates are fast enough to affect the balance between aerosol mass accumulation and loss.

The generality of the PRAD model lies in the choice of physical and chemical constants, such as diffusion coefficients, Henry's law constants, reaction rate coefficients and equilibrium constants. If another system were to be investigated, constants could be replaced appropriately to represent decarboxylation, molecular transport and solubility of another iron-carboxylate

- 95 complex. Another generality is that diffusion coefficients of  $CO_2$  and  $H_2O$  are known with great certainty (Dou et al., 2019). Since citric acid is regarded as an proxy for atmospheric secondary organic aerosol compounds in terms of being highly oxygenated and having a representative viscosity, we have some confidence to expand our system to apply to mass loss of atmospheric particles. Finally, we consider iron re-oxidation rates as reliable (Figs. 8, 10 and 11), as this occurs on the same scales as mass loss rates.
- 100 Model improvements would involve a representation of peroxyl radical chemistry, since photochemical decarboxylation steps of other iron(III)-carboxylate complexes may not immediately result in  $HO_2$  production. By adding more reactions it would make our model more general in its scope, which will be the focus of future work. Additionally, reaction rate coefficients of the two competitive reactions of  $O_2$  with either organic radicals or with iron(II)-carboxylate complexes are not yet well constrained. We have stated at the end of section 2.5 that a different combination of reaction rate coefficients, diffusion
- 105 coefficients and Henry's Law constants may yield a satisfactory representation of our data. See also Appendix A5. Although, our experimental results and model constrains overall loss rates, it cannot constrain parameters individually.

Finally, we note that the radical production predicted with the PRAD model is reliable. Since re-oxidation and mass loss is tied to the production of radicals, then radical production is also a reliable and generalizable model output. This is a focus of another paper currently in review and will only briefly be mentioned in our revised manuscript.

110 Changes to the manuscript: We will split section 3.5 into two paragraphs.

We will begin the section by adding the following sentence.

"The PRAD model was developed to also be used in more general particle systems. After establishing..."

On line 425 we will add the following discussion:

- "To better understand the importance of mass loss in this generalized system, we make a simple and modest comparison of mass loss in Fig. 13 on the order of 20% over 5 hours with ambient mass accumulation measured in the field. Our results are equal to a mass loss rate of about  $0.4 \ \mu g \ m^{-3} (air) \ hr^{-1}$  assuming an aerosol population with an organic mass of ~ 10  $\ \mu g \ m^{-3} (air)$  undergoing iron-carboxylate photochemistry. This is much larger than observed organic mass accumulation in ambient air masses due to photochemical aging during atmospheric transport at about 0.06  $\ \mu g \ m^{-3} \ hr^{-1}$  or 6  $\ \mu g \ m^{-3}$  over 4 days (Zaveri et al., 2012; Moffet et al., 2012). This implies, that the mass loss rates are fast enough to affect the balance between earosol mass accumulation and lose."
- 120 between aerosol mass accumulation and loss."

And we will add the following sentences to the conclusion section:

"Although a systematic study exploring the whole range of atmospheric conditions was beyond the scope to this work, there are some aspects of the PRAD model and certain parameters that we argue are reliable and pertainate to atmospheric

aerosol photochemistry. First, coefficients in the PRAD model framework can be changed to predict mass loss rates of a

- 125 different iron-carboxylate complex system. We are fairly confident that diffusion coefficients of  $CO_2$  and  $H_2O$  can be used for atmospheric aerosol particles as these were obtained in a more targeted study (Dou et al., 2019). Mass loss rates in general are fairly reliable to be used in atmospheric particles as these are linked to photochemical reaction rates which have been characterized (Weller et al., 2013, 2014). Finally, reoxidation rates and production of radicals are also reliable, as the system is largely reacto-diffusion limited (see appendix A4) and these occur on the same scales as observed mass loss rates. In our
- 130 companion paper (Alpert et al., 2020, under review), we show a detailed analysis of radical concentrations in ambient aerosol particles for a range of atmospheric conditions and iron content. However, our model still needs major improvements, such as including peroxyl radical chemistry and better constraints on individual parameters such as diffusion coefficients and reaction rate constants. The overall rate may be well-constrained by our experimental studies, however more targeted observations may be necessary for an accurate representation of O2 chemistry, solubility and molecular transport independently of each other
- 135 within aerosol particles."

2) Some additional discussion about the sensitivity that the manual tuning of certain parameters in the PRAD model has would be useful. Quantifying the sensitivity would be excellent but at least something to help the reader gauge the effect that a slight mistuning might have. In a similar vein, adding a column to table 1 showing the uncertainties in each of these parameters (if they're available) would be good.

140

145

0 This point has been raised by reviewer 2 as well. We refer to our answers related to the sensitivity to those given in reply to reviewer 2. As our experiments do not allow to constrain all parameters of the PRAD model individually, we restrain from giving uncertainties in Table 1 and Table 2.

3) Line 322-323 and Figure 7 shows a statistical agreement with 95% confidence between the model and the data, but it is just on the edge of significance. Even though a 95% confidence level is commonly used, it is ultimately arbitrary. A sentence or two giving some extra context or how the estimate could improve would be helpful in making the conclusion (that jcalc is a

good estimator for jobs) more robust.

We agree with the reviewer that some context would help explain why we made such an estimate of the photochemical reaction rate and help to explain how this could be improved. In Fig. 7 the calculated value of j used in the model yield more reduction than observed. This represents a single UV-fiber setup. In Fig. 8, the fiber was setup an addition 2 times, once for

- 150 experiments at RH=40% and another time for experiments at RH=50-60%. Still, we see that the reduction of iron very soon (minutes) after UV light was switched off is still in agreement with observed values of  $\beta$ . In order to better quantify this uncertainty and any possible systematic errors, we should repeat UV-fiber setup procedures and measurements of  $\beta$ , however, the time to use the X-ray beam allotted to us was highly limited and our estimated uncertainty was already good enough. As recommended by the reviewer, we aill add in a few sentences to explain this.
- 155 Minor comments

*Line 45: "humic like" should be hyphenated Line 57: "time resolved" should be hyphenated* Agreed and changed

*Line 72: The chemical formula of iron(III) citrate is FeC6H507 but the structural formula written shows an incorrect number of Carbon and Hydrogen atoms. Please revise the structural formula.*

160 Agreed and changed to  $Fe^{III}(OOCCH_2)_2C(OH)(COO)$

*Line 99: reword: "The PRAD model allows to simulate"*

Sentence rephrased as: "In addition, we will use the PRAD model to simulate photochemical aging processes under atmospheric conditions."

Line 201: The denominator in the expanded version of equation (4) uses "r" as a subscript, it shows "0.5(rr+1 - ri-1)" 165 should it read "0.5(ri+1 - ri-1)" instead? Also why use 0.5(ri+1 - ri-1) instead of (ri+1 - ri)?

We thank the reviewer for noticing, yes, the equation should read:

$$f_i = -4\pi r_i^2 D_1 \frac{dc}{dr}\Big|_{r=r_i} = -4\pi r_i^2 D_1 \frac{c_{i+1} - c_i}{0.5(r_{i+1} - r_{i-1})}; \qquad \forall i \in \{1, 2, \dots, n-1\},\tag{1}$$

To represent the **average** distance for molecules to be transported between two adjacent shells we use  $0.5(r_{i+1} - r_{i-1})$  instead of  $(r_{i+1} - r_i)$ .

170 *Line 209: Shouldn't the radius should be squared in equation (6) as part of the sphere surface area equation? This continues through equations (8), (9), (13), (14), and (16). A brief clarification would help.*

In the outermost shell n, the flux into the gas phase is given by:

$$f_{i} = -4\pi r_{n}^{2} D_{d} \frac{dc}{dr} \bigg|_{r=r_{n}} = -4\pi r_{n}^{2} D_{g} \frac{c_{g} - c_{g}^{*}}{r_{n}} = -4\pi r_{n}^{2} D_{g} \frac{p_{partial} - p_{vapor}}{r_{n} RT};$$
(2)

To clarify we will change eq. (6) accordingly.

175 *Line* 248: If cn is molar concentration (as defined in line 205), then Nn should be the number of moles, not the number of molecules.

We thank the reviewer for pointing this out. We will define  $N_n$  as the moles of molecules in the outermost shell to ensure consistence.

*Line 270: Because concentration was defined as a subscripted "c" (i.e. cn in line 205) replace the nCit values with cCit for consistency.*

We will change  $n_{Cit}^*$ ,  $n_{Cit}$  and  $n_{FeCit}$  to  $N_{Cit}^*$ ,  $N_{Cit}$  and  $N_{FeCit}$  to represent the moles of each species.

Figure 6: The caption colors do not correspond to the figure colors.

We thank the reviewer for pointing out this mistake. The color descriptions in the caption have been corrected.

Line 304: "The degradation progresses..." should be "The degradation processes..."

185 Agreed and changed

**3 Reviewer 4 comments**

My main critique involves how applicable this system is issues of a global scale, as the paper attempts to address in the conclusions. How important are organic-Fe particles globally? Are they so abundant that they are expected to measurably affect the organic aerosol burden? How representative is the Fe citrate system of globally distributed aerosol iron?

- In response to the reviewer comments, we would like to include more details in the introduction and discussion section about the global relevance of iron-carboxylate complexes, in general, and the applicability of the specific iron(III)-citrate system to atmospheric iron-carboxlate compounds. First, organic-Fe particles impact many global processes, such as, determining iron solubility in the atmosphere and deposition in the oceans (Hamilton et al., 2019). Second, auto-oxidation of  $SO_2$  is a wellknown chemical process in fog and cloud water as the result of the interaction of carboxylates and iron to produce sulfate
- 195 and impact aerosol inorganic mass (Grgić et al., 1998, 1999; Grgič, 2009). Additionally, iron photochemical cycling produces a significant amount of radicals (Fang et al., 2020) that can subsequently react with other organics in particles, fog droplets and clouds water affecting the fomation of aqueous phase secondary organic aerosol (aqSOA) (Bianco et al., 2020). Also, photochemical reactions with iron-organic complexes should not be neglected when considering the loss of (aqSOA) in aerosol particles (Weller et al., 2014; Herrmann et al., 2015). Finally, researchers have only shown recently that soluble iron is largely
- 200 complexed with carboxylates in ambient aerosol particles (Tapparo et al., 2020; Tao and Murphy, 2019). Iron is certainly abundant in the atmosphere, however, a global estimate and uncertainty of how iron-organic complexes affect the organic aerosol burden has not yet been realized. The first photochemical model of iron carboxylate complexes and organic aerosol mass loss, to our knowledge, is present in our manuscript. We encourage future work to quantify this effect.
- Atmospheric iron(III) carboxylate photochemistry has been established as important using a slew of different carboxylate compounds (Weller et al., 2014, 2013; Herrmann et al., 2015). All iron complexes with carboxalates, such as pyruvate, glyoxalate, malonate, oxalate, succinate, tartronate, tartrate and citrate undergo ligidand-to-metal charge transfer and decarboxylation (Cieśla et al., 2004; Weller et al., 2013, 2014). Furthermore, citric acid is regarded as an excellent proxy for atmospheric secondary organic aerosol compound in terms of being highly oxygenated and having a similar viscosity (Lienhard et al., 2015; Reid et al., 2018). Therefore, it is highly justified to claim that iron(III)-citrate photochemistry is a system relevant to
- 210 atmospheric aerosol particles and cloud droplets.

We will include this discussion on lines 73-76 with the following text.

"Our FeIII(Cit) system undergoes LMCT reaction in the same way as countless other iron(III)-carboxylate compounds (Cieśla et al., 2004; Weller et al., 2013, 2014). Its photochemical reaction scheme is well established... water diffusivity and viscosity being well studied (Lienhard et al., 2012, 2014, 2015; Song et al., 2016). For these reasons, it is a valid and reliable proxy for atmospheric iron-carboxylate photochemical processes."

On line 44, we will also discuss the global importance more effectively.

"Quantifying iron atmospheric processing and solubility is of global importance, especially for nutrient input into the Worlds oceans (Hamilton et al., 2019; Kanakidou et al., 2018). Heterogeneous chemistry involving particulate iron and  $SO_2$  can result in sulfate formation and increase aerosol loading (Grgić et al., 1998, 1999; Grgič, 2009). Additionally, iron photochemical

220 processing in aerosol particles, fog droplets and cloud water is an important radical source (Bianco et al., 2020; Abida et al., 2012) and sink for organic compounds (Weller et al., 2014, 2013; Herrmann et al., 2015)." And we will add to the end of the paragraph: "Field studies have confirmed that soluble iron is mostly in complexes with carboxylate functions (Tapparo et al., 2020; Tao and Murphy, 2019)."

My specific comments are as follows:

225

Line 20: I suggest adding a brief description of the type of parameters used to "tune" the model.

This comment was also asked by other reviewers, and as a result we have added new text to detail the tuning of the PRAD model. Briefly, we marked the equilibrium constants and rate coefficients which were tuned in Table 2 and added detailed text on tuning parameters at the end of section 2.5. Briefly, all parameters were adjusted in a wide and acceptable range until a good representation of our data could be obtained. Tuning of individual bulk diffusion coefficients for all species was highly

230 impractical and therefore a parameterization was made as a function of molar mass in appendix A1. Parameterization constants were tuned resulting in the absolute diffusion coefficients shown in Fig. A1. Henry's law coefficients for gasses were tuned, however purposefully set at values higher than expected for pure water or highly dilute aqueous solution. This was inspired by

previous studies regularly reporting solubility of e.g.  $O_2$  and  $CO_2$  higher in a variety organic liquids than water (Fogg, 1992; Battino et al., 1983).

235 Line 119: It is stated that refractive index is assumed not to change. Shouldn't the refractive index change with the amount of water on the particle? If the RH changes, so will the refractive index. Was this accounted for? Refractive indices for both dry and aqueous phases should be available and used to show this is a good assumption. If its not a good assumption, how might it affect results and conclusions?

We completely agree with the reviewer that any compositional change will lead to a change in the refractive index. However

- 240 note, that all our EDB experiments have been at constant RH and temperature, so that compositional change occurs through the photochemical reactions. The data shown in Fig. 2(d) allow us to observe this effect. If density and refractive index of the particle would indeed stay constant during photochemistry, the mass to initial mass ratio obtained by the mass measurement (compensation of gravitational force by the DC-voltage) should agree with the one obtained from sizing. Up to a mass loss of about 20% this seems to be an excellent approximation as the two curves almost overlap. Later they deviate from each
- 245 other with progressing chemical evolution, indicative of a change in density and refractive index. This means that initially the refractive index and density is strongly dominated by that of the major compound, citric acid. Therefore, we restrict ourselves to the initial mass loss (about 20%, when comparing model simulations and experiment later, cp. Fig. 6 and corresponding discussion.

*Line 125: Molar absorptivities are stated, and will affect reaction rates. How likely is it for side products to be formed that have different optical properties?*

We agree with the reviewer that the side products participating in the photochemistry (given as R10 to R14 in Table 2) will have different molar absorptivities or absorption cross sections. As can be seen in the Table 2 we take those as unknown tuning parameters. At least partly, these products may be responsible for the observed acceleration of the photochemical degradation. However, as our chemical scheme neglects other products like peroxyl radicals and we do not have any detailed information on

255 these compounds but rather treat them in a lumped manner, we cannot deduce any reliable cross sections for these compounds. *Line 126: I find the term "O2 gas phase" and "oxygen atmosphere" ambiguous and confusing. Are these atmospheres of pure O2? I think it would be best to spell this out.*

Yes, they both refer to pure  $O_2$ . To avoid confusion we will write "pure  $O_2$  gas phase" in the revised manuscript. *Line 190: Change "devides" to "divides"*

260 Typo corrected

250

265

Line 329: Change "dryer" to "drier"

Typo corrected

Figure 13: How representative are these sizes of Fe containing particles? Also, if it is dust, how applicable are the results obtained on the iron citrate system which is homogeneously mixed and all soluble? Also, panel B is for a 200 nm diameter particle. Why not increase the particle size for a more reasonable Fe containing particle size?

We believe to cover the important size range in Fig. 13(a) for iron containing particles (100 nm to 2  $\mu$ m), see Moffet et al. (2012). While larger particles may contain a larger fraction of iron, there is clear evidence that particles of 200 nm size contain iron. Larger particles will need longer times for reaching the same relative mass loss compared to smaller particles, but panel (b) is meant to illustrate the humidity dependence.

270 Line 425: How applicable are the EDM experiments to smaller particles? Is it possible to have a size dependent photolysis rate for reasons beyond that described in the PRAD model? What about small particle optical effects?

The reviewer raises an interesting question. For example nano-focusing could conceptually change photochemical rates. We were not able to study such effects in our setup and could only speculate here. However, taking all other uncertainties into account we are convinced that such effects are of secondary importance. But we agree with the reviewer that experiments

275 looking into such effects are desirable to be performed in the future.

**References**

[revised manuscript text omitted]